# GRADIENT-BASED TUNING OF HAMILTONIAN MONTE CARLO HYPERPARAMETERS

## ABSTRACT

Hamiltonian Monte Carlo (HMC) is one of the most successful sampling methods in machine learning. However, its performance is significantly affected by the choice of hyperparameter values, which require careful tuning. Existing approaches for automating this task either optimize a proxy for mixing speed or consider the HMC chain as an implicit variational distribution and optimize a tractable lower bound that is too loose to be useful in practice. Instead, we propose to optimize an objective that quantifies directly the speed of convergence to the target distribution. Our objective can be easily optimized using stochastic gradient descent. We evaluate our proposed method and compare to baselines on a variety of problems including synthetic 2D distributions, the posteriors of variational autoencoders and the Boltzmann distribution for molecular configurations of a 22 atom molecule. We find our method is competitive with or improves upon alternative baselines on all problems we consider.

## 1 INTRODUCTION

Hamiltonian Monte Carlo (HMC) is a popular sampling based method for performing accurate inference on complex distributions that we may only know up to a normalization constant (Neal, 2011). Unfortunately, HMC can be slow to run in practice as we need to allow time for the simulation to 'burn-in' and also to sufficiently explore the full extent of the target distribution. Tuning the HMC hyperparameters can help alleviate these issues but this requires domain expertise and must be repeated for every problem HMC is applied to.

There have been many attempts to provide an automatic method of tuning the hyperparameters. Some methods use a proxy for the mixing speed of the chain, i.e. the speed at which the Markov chain marginal distribution approaches the target. For example, Levy et al. (2018) use a variation on the expected squared jumped distance to tune parameters in order to encourage the chain to make large moves within the sample space. Other methods draw upon ideas from Variational Inference (VI). VI (Jordan et al., 1999) is an optimization based method that is often contrasted to Markov Chain Monte Carlo methods such as HMC. In VI, we approximate the target using a parametric distribution, reducing the approximation bias through optimizing the distribution parameters. The optimization procedure maximises a lower bound on the normalization constant of the target which is equivalent to minimising the KL-divergence between the approximation and the target. To apply this idea to HMC, Salimans et al. (2015); Wolf et al. (2016) consider the marginal distribution of the final state in a finite length HMC chain as an implicit variational distribution with the intention of tuning the HMC parameters using the VI approach. However, the implicit distribution makes the usual variational lower bound intractable. To restore tractability, they make the bound looser by introducing an auxiliary inference distribution approximating the reverse dynamics of the chain. The looseness of the bound depends on the KL-divergence between the auxiliary inference distribution and the true reverse dynamics. As the chain length increases, the dimensionality of these distributions increases, tending to increase the looseness of the bound. This causes issues during optimization because the increasing magnitude of this extra KL-divergence term encourages the model to fit to the imperfect auxiliary inference distribution as opposed to the target as desired. Indeed, Salimans et al. (2015) only consider very short HMC chains using their method.

In this work, we further investigate the combined VI-HMC approach as this has the potential to provide a direct measure of the chain's convergence without the need to rely on proxies for per-

formance. When applied to an implicit HMC marginal distribution, the variational objective can be broken down into the tractable expectation of the log target density and the intractable entropy of the variational approximation. This entropy term prevents a fully flexible variational distribution from collapsing to a point mass maximizing the log target density. Since HMC, by construction, cannot collapse to such a point mass, we argue that the entropy term can be dropped provided the initial distribution of the chain has enough coverage of the target. We evaluate our proposed method on a variety of tasks. We first consider a range of synthetic 2D distributions before moving on to higher dimensional problems. In particular, we use our method to train deep latent variable models on the MNIST and FashionMNIST datasets. We also evaluate on a popular statistical mechanics benchmark: sampling molecular configurations from the Boltzmann distribution of a 22 atom molecule, Alanine Dipeptide. Our results show that this method is competitive with or can improve upon alternative tuning methods for HMC on all problems we consider.

## 2 BACKGROUND

### 2.1 HAMILTONIAN MONTE CARLO

HMC is a Markov Chain Monte Carlo method (Neal, 1993) which aims to draw samples from the $n$-dimensional target distribution $p(x) = \frac{1}{Z} p^*(x)$ where $Z$ is the (usually unknown) normalization constant. It introduces an auxiliary variable $\nu \in \mathbb{R}^n$, referred to as the momentum, which is distributed according to $\mathcal{N}(\nu; \mathbf{0}, \text{diag}(\boldsymbol{m}))$, with the resulting method sampling on the extended space $\zeta = (x, \nu)$. HMC progresses by first sampling an initial state from some initial distribution and then iteratively proposing new states and accepting/rejecting them according to an acceptance probability. To propose a new state, first, a new value for the momentum is drawn from $\mathcal{N}(\nu; \mathbf{0}, \text{diag}(\boldsymbol{m}))$, then, we simulate Hamiltonian Dynamics with Hamiltonian, $H(x, \nu) = -\log p^*(x) + \frac{1}{2} \nu^T \text{diag}(\boldsymbol{m})^{-1} \nu$ arriving at new state $(x', \nu')$. This new state is accepted with probability $\min[1, \exp(-H(x', \nu') + H(x, \nu))]$ otherwise we reject the proposed state and remain at the starting state. The Hamiltonian Dynamics are simulated using a numerical integrator, the leapfrog integrator (Hairer et al., 2003) being a popular choice. To propose the new state, $L$ leapfrog updates are taken, each update consisting of the following equations:

$$\nu_{k+\frac{1}{2}} = \nu_k + \frac{1}{2} \boldsymbol{\epsilon} \odot \nabla_{x_k} \log p^*(x_k)$$

$$x_{k+1} = x_k + \nu_{k+\frac{1}{2}} \odot \boldsymbol{\epsilon} \odot \frac{1}{\boldsymbol{m}}$$

$$\nu_{k+1} = \nu_{k+\frac{1}{2}} + \frac{1}{2} \boldsymbol{\epsilon} \odot \nabla_{x_{k+1}} \log p^*(x_{k+1})$$

where $\frac{1}{\boldsymbol{m}} = (\frac{1}{m_1}, \ldots, \frac{1}{m_n})$ and $\odot$ denotes element wise multiplication. The step size, $\boldsymbol{\epsilon}$, and the mass, $\boldsymbol{m}$, are hyperparameters that need to be tuned for each problem the method is applied to. We note that in the usual definition of HMC, a single scalar valued $\epsilon$ is used. Our use of a vector $\boldsymbol{\epsilon}$ implies a different step size in each dimension which, with proper tuning, can improve performance by taking into account the different scales in each dimension. The use of $\boldsymbol{\epsilon}$ does mean the procedure can no longer be interpreted as simulating Hamiltonian Dynamics, however, it can still be used as a valid HMC proposal (Neal, 2011). We do not consider the problem of choosing $L$ in this work.

### 2.2 VARIATIONAL INFERENCE

VI approximates the target $p(x)$ with a tractable distribution $q_\phi(x)$ parameterized by $\phi$. We choose $\phi$ as to minimise the Kullback-Leibler divergence with the target, $D_{KL}(q_\phi(x) || p(x))$. As we only know $p(x)$ up to a normalization constant, we can equivalently choose $\phi$ as to maximise the tractable Evidence Lower-Bound (ELBO):

$$\text{ELBO} = \log Z - D_{KL}(q_\phi(x) || p(x)) = \mathbb{E}_{q_\phi(x)}[\log p^*(x) - \log q_\phi(x)].$$

## 3 HYPERPARAMETER TUNING THROUGH THE EXPECTED LOG-TARGET

VI tunes the parameters of an approximate distribution to make it closer to the target. We would like to use this idea to tune the hyperparameters of HMC. We can run multiple parallel HMC chains and

treat the final states as independent samples from an implicit variational distribution. If each chain starts at initial distribution $q^{(0)}(x)$ and then runs $T$ accept/reject cycles, we can denote this implicit distribution as $q_\phi^{(T)}(x)$, where $\phi$ now represents the HMC hyperparameters. Ideally, we would then choose $\phi$ as to maximise the ELBO:

$$\phi^* = \underset{\phi}{\operatorname{argmax}} \, \mathbb{E}_{q_\phi^{(T)}(x)}\big[\log p^*(x) - \log q_\phi^{(T)}(x)\big] = \underset{\phi}{\operatorname{argmax}} \, \mathbb{E}_{q_\phi^{(T)}(x)}\big[\log p^*(x)\big] + H\left(q_\phi^{(T)}(x)\right).$$

Whilst the first term in this expression can be estimated directly via Monte Carlo, the entropy term, $H\big(q_\phi^{(T)}(x)\big)$, is intractable. To get around this, we should consider the purpose of the two terms during optimization. Maximizing the first term encourages $q_\phi^{(T)}(x)$ to produce samples that are in the high probability regions of the target, i.e., ensuring that $q_\phi^{(T)}(x)$ is high where $\log p^*(x)$ is high. The entropy term acts as a regularizer preventing $q_\phi^{(T)}(x)$ from simply collapsing to a delta function at the mode of $p(x)$. The key observation of our method is that HMC already fulfills this regularization role because the implicit distribution it defines is not fully flexible. If $q_\phi^{(T)}(x)$ were to collapse to a delta function, this would require the hyperparameters to be such that the HMC scheme guides samples to the same point in space no matter their starting position, as sampled from $q^{(0)}(x)$, which is unreasonable for practical problems. Therefore, we propose tuning $\phi$ simply by maximizing the expected log target density under the final state of the chain:

$$\phi^* = \underset{\phi}{\operatorname{argmax}} \, \mathbb{E}_{q_\phi^{(T)}(x)}\big[\log p^*(x)\big]. \tag{1}$$

Although HMC does have a regularization effect, removing the entropy term does have some implications that we need to consider. Namely, if the initial distribution, $q^{(0)}(x)$, is concentrated in a very high probability region of the target, $p(x)$, then optimizing objective (1) will not encourage HMC to explore the full target. Conversely, it would encourage the chains to remain in this region of high probability, close to their initial sampling point, which is undesirable behaviour. The key to avoiding this problem is to choose an initial distribution that has a sufficiently wide coverage of the target, we will discuss methods for doing this in Section 4.

We perform the optimization in (1) using stochastic gradient ascent with gradients being obtained using the standard reparameterization trick (Rezende et al., 2014; Kingma & Welling, 2014). Samples from $q_\phi^{(T)}(x)$ can be obtained through a deterministic function of primitive random variables: $x_0 \sim q^{(0)}(x)$ for the initial distribution, $\gamma_{0:T-1}, \gamma_t \sim \mathcal{N}(0, \boldsymbol{I})$ for the momentum variables and $a_{0:T-1}, a_t \sim \mathcal{U}[0, 1]$ for the accept/reject decisions. We denote this function as $f_\phi(x_0, \gamma_{0:T-1}, a_{0:T-1})$. For minibatches of size $N$, the gradient is then estimated by

$$\frac{1}{N} \sum_{n=1}^{N} \nabla_\phi \log p^* \left( f_\phi \left( x_0^{(n)}, \gamma_{0:T-1}^{(n)}, a_{0:T-1}^{(n)} \right) \right). \tag{2}$$

Each gradient value can be calculated using automatic differentiation tools. For (2) to be strictly unbiased, $f_\phi$ must obey certain smoothness constraints to allow differentiation under the integral sign (Border, 2016). This does not hold in this case due to the accept/reject step. Other works have found success in making this approximation (Levy et al., 2018; Thin et al., 2020) and as Thin et al. (2020) point out, this same issue is encountered when using the very popular ReLU activation function. Our empirical results confirm that (2) enables effective optimization of (1). We also make another approximation when using automatic differentiation. For the leapfrog updates, we stop the backpropagation of the gradient through $x_k$ in $\nabla_{x_k} \log p^*(x_k)$ to prevent the calculation of second-order gradients. We find this has little impact on the convergence of the algorithm and can lead to $5\times$ speedups in execution time.

### 3.1 Demonstration on a toy problem

We now demonstrate the ideas of the previous section on a very simple toy problem. Here the target is a 1-dimensional normal distribution $\mathcal{N}(0, 1)$ which we attempt to sample from using a 10 step HMC chain with each step consisting of 5 leapfrog updates. We initialize the chain either with a narrow initial distribution $\mathcal{N}(0, 0.25)$ or a wide initial distribution $\mathcal{N}(0, 4)$. We keep $\boldsymbol{m}$ constant for all steps but train one step size $\boldsymbol{\epsilon}_t$ for each HMC step. Figure 1 plots the progression of $\mathbb{E}_{q_\phi^{(t)}}\big[\log p(x)\big]$ during sampling for these two cases, before and after training. Before training,

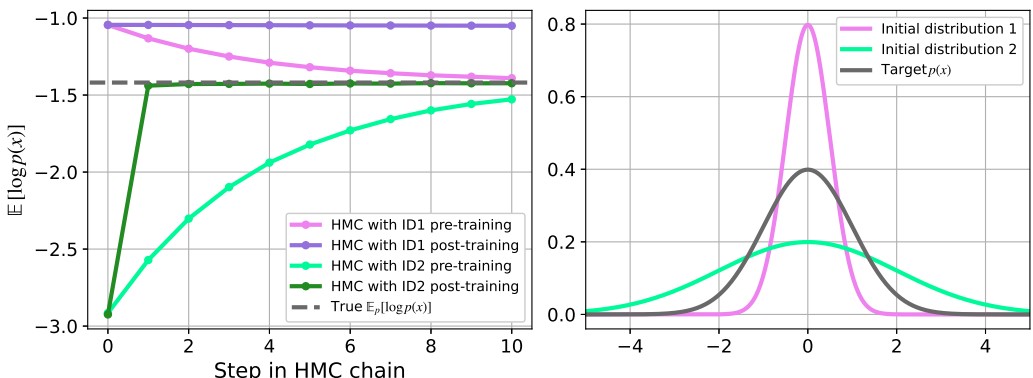

Figure 1: (Left) $\mathbb{E}_{q_\phi^{(t)}}\left[\log p(x)\right]$ as a function of step $t$ in the HMC chain for initial distribution 1, $\mathcal{N}(0, 0.25)$, and initial distribution 2, $\mathcal{N}(0, 4)$, before and after training. The 'true' value, $\mathbb{E}_p\left[\log p(x)\right]$ is also plotted. (Right) The target pdf along with the two initial distributions.

when using the narrow initial distribution, $\mathbb{E}_{q_\phi^{(t)}}\left[\log p(x)\right]$ initially starts above the true value but converges from above as the marginal HMC distribution, $q_\phi^{(t)}$, spreads to cover the target. However, after training by optimizing eq (1), all the step sizes have become very small causing the HMC chains to remain at their initial sampled positions which is obviously detrimental for convergence. To avoid this, we can use a wide initial distribution. Figure 1 shows that, in this case, optimizing (1) greatly speeds up convergence to the true distribution. We give the code for all our experiments in the supplementary material.

## 4 METHODS FOR FINDING A SUITABLE INITIAL DISTRIBUTION

In order for the method to be useful, we need to use an initial distribution that sufficiently covers the mass of the target. On the other hand, it would be unhelpful to use a distribution that was overly spread out relative to the target because we are focusing on short parallel HMC chains so we would like to keep the burn-in time to a minimum. Although ultimately this trade-off is an engineering problem, we provide an automatic method in this section that we have empirically evaluated on a variety of applications and have found to give consistently good results. The main idea is to use a variational approximation to the target as the initial distribution, as done by Hoffman (2017). This should be a distribution that can be easily sampled from and easily tuned to fit the target, e.g. a Gaussian distribution or a normalizing flow (Tabak & Vanden-Eijnden, 2010; Rezende & Mohamed, 2015). Rather than use the standard ELBO, we use $\alpha$-divergence minimization (Hernández-Lobato et al., 2016) for training. The $\alpha$ value dictates the mass covering behaviour of the resulting approximation, with $\alpha = 0$ corresponding to the standard mode seeking $D_{KL}\left(q_\phi(x)||p(x)\right)$ minimization and $\alpha = 1$ corresponding to the mass covering $D_{KL}\left(p(x)||q_\phi(x)\right)$ minimization. We compare both $\alpha$ values in our experiments. The $\alpha$-divergence is very useful in this context as it can provide us with a mass covering approximation without the use of samples from the target, it requires only the (unnormalized) target density, regardless of the value of $\alpha$. However, if samples from the target are available, then alternatively $q_\phi(x)$ can be tuned via maximum likelihood which is also mass-covering.

Note that the previous approaches will produce an initial distribution that fits the target, but do not guarantee that this initial distribution will be broad enough. To address this, we also allow our method to automatically adjust the width of the initial distribution as necessary to keep $q_\phi^{(T)}(x)$ as closely matched to $p(x)$ as possible. This is achieved by applying a scalar scale factor $s$ centered around the mean $\mu$ to each sample $x_i$ from the initial distribution, i.e. using $\hat{x}_i = s(x_i - \mu) + \mu$ as our sample. In particular, we train $s$ by minimizing the Sliced Kernelized Stein Discrepancy (Gong et al., 2020) or SKSD[1] between the final state distribution and the target, $\text{SKSD}(q_\phi^{(T)}(x), p(x))$.

---

[1] We use the metric referred to as the maxSKSD in Gong et al. (2020)

Table 1: KSD between the HMC samples and the target distribution for the baselines and the 4 variations of our method on each of the synthetic target distributions.

|  | Gaussian | Laplace | Dual Moon | Mixture | Wave 1 | Wave 2 | Wave 3 |
|---|---|---|---|---|---|---|---|
| $\alpha = 0$ | 0.0677 | 0.0005 | 0.2370 | 0.0004 | 0.0224 | 0.0525 | 0.0462 |
| $\alpha = 1$ | 0.0009 | **0.0004** | 0.8637 | 0.0010 | 0.0067 | 0.0158 | 0.0801 |
| SKSD & $\alpha = 0$ | **0.0008** | 0.0016 | **0.1684** | **0.0004** | **0.0017** | 0.0020 | **0.0217** |
| SKSD & $\alpha = 1$ | 0.0009 | 0.0014 | 0.2528 | 0.0004 | 0.0017 | **0.0019** | 0.0317 |
| $\min \bar{p} = 0.25$ | 0.0364 | 0.0005 | 1.1553 | 0.0846 | 0.0645 | 0.9447 | 0.0465 |
| NUTS | 0.0044 | 0.0016 | 0.2326 | 0.0023 | 0.0124 | 0.0260 | 0.0965 |

The SKSD is a differentiable scalable discrepancy measure requiring only samples from $q_\phi^{(T)}$ and gradients of the target, $\nabla_x \log p^*(x)$. This objective encourages suitable values of $s$ because, say $s$ is too small, then the $\phi$ training (which occurs jointly with $s$ training) will result in a degenerate $q_\phi^{(T)}(x)$ far from the target. The SKSD measures this discrepancy and provides a learning signal for increasing $s$. Conversely, if $s$ is too large then $q_\phi^{(T)}(x)$ will also be far from the target since the chain will not be able to account for the extremely poor initialization. The SKSD will then result in a learning signal to decrease $s$ to a more reasonable value. Given that the SKSD is a tractable objective that measures the discrepancy between $q_\phi^{(T)}(x)$ and $p(x)$, it is theoretically feasible to use the SKSD to optimize $\phi$ too, but we found that the SKSD does not perform well when optimizing too many parameters, which is why (1) is used instead. Note, however, that the SKSD works very well in practice when we only tune the single scalar parameter $s$.

## 5 EXPERIMENTS

### 5.1 2D DISTRIBUTIONS

We evaluate our tuning method on a range of synthetic 2D target densities shown in Figure 2a. The equations for these distributions are listed in Appendix A.1. We use 30-step HMC chains initialized with a factorized Gaussian approximation, which is trained by minimizing the $\alpha$-divergence. We consider 4 variations of our method in which we optimize individual step sizes and masses for each dimension and HMC step using (1) but vary the value of $\alpha \in \{0, 1\}$ and whether or not to include the tuning of the scaling $s$ by minimizing the SKSD ($s = 1$ when not tuned). We include two baselines for reference. The first one is taken from Hoffman (2017), where the step size in each dimension $k$ is given by $\sigma_k \epsilon_0$ with $\sigma_k$ being the standard deviation in dimension $k$, as estimated by a Gaussian fitted by minimizing the $\alpha = 0$-divergence and $\epsilon_0$ being adjusted as to control the minimum acceptance probability over a batch of parallel chains. In line with Hoffman (2017) we set this minimum acceptance probability to 0.25. The second baseline is the popular No-U-Turn Sampler (Hoffman & Gelman, 2011)[2]. We use the dual averaging variant of the No-U-Turn Sampler, which includes a method for tuning the step size as to encourage an equivalent of the HMC average acceptance probability towards a target $\delta \in (0, 1)$. We set $\delta = 0.2$ in our experiments. To quantify convergence to the target, we used the Kernelized Stein Discrepancy (KSD) (Liu et al., 2016; Chwialkowski et al., 2016) between the generated samples and the targets. The results are shown in Table 1. Our method consistently outperforms the baselines. Furthermore, the methods that tune $s$ by minimizing the SKSD tend to fit the targets better than those methods that do not tune this scale factor. For some distributions, the scale factor effectively helps prevent mode seeking. In Appendix A.2, we confirm this quantitatively by comparing expected log target values and find that narrow initial distributions ($\alpha = 0$) often lead to artificially high values with the SKSD preventing this pathology.

### 5.2 DEEP LATENT GAUSSIAN MODELS

We now use our method to train Deep Latent Gaussian Models or DLGMs (Kingma & Welling, 2014; Rezende et al., 2014). DLGMs are popular generative models that describe observed data $x$ by the following generative process: first sample a latent variable $z \sim \mathcal{N}(0, \boldsymbol{I})$ and then sample $x$

---

[2]We use the implementation from `https://github.com/mfouesneau/NUTS` in our experiments.

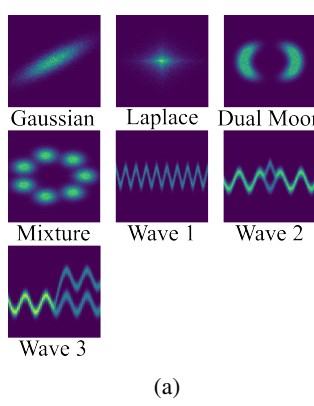

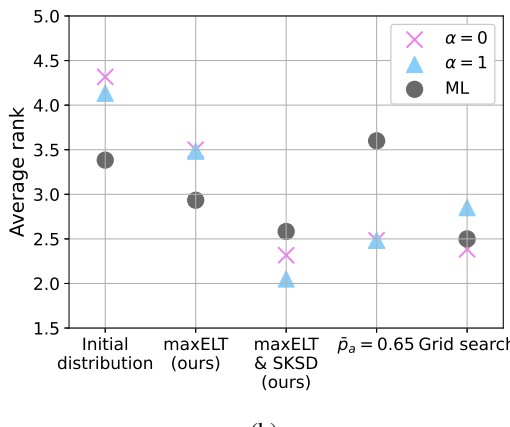

(a)

(b)

Figure 2: (a) Histograms of 2D targets generated by rejection sampling. (b) Average ranking of marginal KL-divergences for each method in the Alanine Dipeptide experiment (lower is better).

from some distribution parameterized by the output of a deep neural network that receives $z$ as input, i.e., $x \sim p_\theta(x|z)$, with $\theta$ being the neural network parameters. The Variational Auto-Encoder (VAE) is a popular method for training these models. This method works by fitting a factorized Gaussian approximation $q_\psi(z|x)$ to the posterior $p_\theta(z|x)$. The means and variance parameters of $q_\psi(z|x)$ are given by a deep neural network that receives $x$ as input. The parameters $\theta$ and $\psi$ are then optimized jointly by maximising an ELBO objective. In our experiments, we consider the HMC sampling distribution, $q_\phi^{(T)}(z|x)$, as a variational approximation with the initial distribution being a factorized Gaussian distribution $q_\psi^{(0)}(z|x)$ whose parameters are given by a deep neural network receiving $x$ as input. The parameters $\psi$ are trained by minimizing an $\alpha$-divergence. We train $(\phi, \theta)$ by maximizing the objective given by (1). Note that we train only one set of HMC hyperparameters for all $x$ values, i.e., we assume the same hyperparameter values will work on all $p_\theta(z|x)$ targets, independently of $x$. One could make the hyperparameters depend on $x$ through an amortization network, but we found that this did not improve performance. We also include a scaling of $q_\psi^{(0)}(z|x)$ with scale factor $s$ trained by minimizing the SKSD. We consider two benchmark datasets: MNIST and Fashion MNIST. As it is common in the literature, we use binarized versions of these datasets, with $p_\theta(x|z)$ giving the parameters of a Bernoulli distribution over the pixels. The likelihood is parameterized with the same convolutional architecture used by Salimans et al. (2015). Full experiment details can be found in Appendix B.1. In these experiments, we found that an appropriate value of $s$ can be successfully found by minimizing the SKSD. The values of $s$ obtained are large enough to prevent our HMC tuning method to overfit to regions of high posterior density, but also small enough to ensure the generation of accurate samples. More details can be found in Appendix B.2.

We evaluate the quality of the models trained with our method on both the MNIST and Fashion MNIST datasets. For each model, we estimate the data marginal log-likelihood, $\log p_\theta(x)$, on each of the test images using Hamiltonian Annealed Importance Sampling or HAIS (Sohl-Dickstein & Culpepper, 2012) and report the average value and its standard error in Table 2. For comparison, we also report log-likelihood values for multiple other methods. Using the same neural architecture, we implemented the standard VAE and IWAE[3] models. We also implemented another method of tuning $\phi$ (Hoffman, 2017) where the step sizes are adjusted to make the minimum average acceptance probability across a batch of images equal to 0.25. We update $\theta$ as in our method and use the same number of leapfrog steps for a fair comparison. Finally, we report the best log-likelihood values from Salimans et al. (2015); Caterini et al. (2018) which include HMC hyperparameter tuning during training and use the same network architecture as us (note that these authors only evaluated on the MNIST dataset). We confirm the difference in average log-likelihood between models is significant by performing a paired t-test for each pairing of models, we report the results in Appendix B.3. We find that on both MNIST and Fashion MNIST, the HMC based methods generally achieve

---

[3]We used the DReG estimator from Tucker et al. (2019) for the IWAE.

Table 2: Average test set marginal log-likelihood and its standard error for different models for MNIST and Fashion MNIST estimated using HAIS. For models with a scale factor, we also report the final scale after training. For comparison, we report the best log-likelihood values from previous works that tune HMC hyperparameters and use the same architecture.

| Model | | MNIST | | | Fashion MNIST | |
| | Scale | Mean | Standard Error | Scale | Mean | Standard Error |
| --- | --- | --- | --- | --- | --- | --- |
| VAE | - | -85.08 | 0.2172 | - | -108.54 | 0.6010 |
| DReG-IWAE | - | -83.73 | 0.2109 | - | -104.48 | 0.5841 |
| $\alpha = 0$ | 1.0 | -83.48 | 0.2101 | 1.0 | -104.08 | 0.5834 |
| $\alpha = 1$ | 1.0 | -82.46 | 0.2073 | 1.0 | -103.57 | 0.5826 |
| $\alpha = 0$ & SKSD | 6.79 | -81.91 | 0.2042 | 5.58 | -103.18 | 0.5802 |
| $\alpha = 1$ & SKSD | 3.90 | -81.94 | 0.2045 | 3.59 | **-102.29** | 0.5748 |
| Hoffman (2017) | - | **-81.74** | 0.2046 | - | -103.04 | 0.5804 |
| Salimans et al. (2015) | - | -81.94 | - | | | |
| Caterini et al. (2018) | - | -82.62 | - | | | |

significantly better performance than VAE or IWAE, showing that reducing the approximation bias of the variational distribution with HMC greatly helps model learning. Furthermore, we see that adding the scale factor to our model significantly improves performance as this avoids degenerate behaviour when training the HMC hyperparameters. We note that, without any scaling, $\alpha = 1$ outperforms $\alpha = 0$ due to $\alpha = 0$ resulting in a too narrow initial distribution. With scaling, the SKSD automatically widens the initial distribution making the performance between the $\alpha$ values similar. Finally, we observe that HMC based methods top out at similar log-likelihood values (within around one standard error), we believe this is due to reaching the limits of the architecture on these datasets with no more gains to be made from more accurate posterior approximations.

## 5.3 MOLECULAR CONFIGURATIONS

For our final experiment, we evaluate our method on the complex real-world problem of sampling equilibrium molecular configurations from the Boltzmann distribution of the molecule Alanine Dipeptide. The unnormalized target distribution given the atom coordinates $x$ is $e^{-u(x)}$, where $u$ is the potential energy of the system, which can be obtained using the laws of physics. While this problem is usually tackled through Molecular Dynamics (MD) simulations, giving a sequence of highly correlated samples, we aim to produce samples using our trained short HMC chains. We do not operate directly on the Cartesian coordinates but apply the coordinate transform presented by Noé et al. (2019), see also Appendix C.2, to map some of the Cartesian coordinates to bond lengths, bond angles, and dihedral angles giving a dimensionality of 60. Our methods use a normalizing flow based on real-valued non-volume preserving (RNVP) transformations (Dinh et al., 2017) as initial distribution, followed by 50 HMC steps. We used various methods to train the initial distribution and HMC hyperparameters (using individual masses and step sizes for each dimension and HMC step in this case). The RNVP models were trained with $\alpha = \{0, 1\}$-divergence and maximum likelihood. The latter was done using $10^5$ training data samples obtained via a MD simulation[4]. In these experiments, we first trained the initial sampling distributions (via ML or $\alpha$-divergence) and then kept them fixed when tuning the HMC hyperparameters and the scale factor. To evaluate the performance of the different methods, a new MD simulation was run to obtain $10^6$ test samples. Since we the model likelihood is intractable, we approximated the 60 marginal distributions by kernel density estimates from the MD test samples and compared them using the KL-divergence with the corresponding density estimates for the samples of the HMC tuning methods. This performance measure also has practical relevance as the marginals of proteins, especially those of the dihedral angles, determine important material properties such as how the protein folds (Ramachandran et al., 1963). The HMC hyperparameters were tuned by maximising the expected log-target without adjusting the initial distribution (referred to as maxELT) or by maximising the expected log-target whilst also tuning the scale factor of the initial distribution by minimizing the SKSD (referred to as maxELT & SKSD). As a baseline, we also optimized the HMC parameters via a grid search, keeping step sizes

---

[4]The same dataset was used to obtain the mean and variances required in the normalization step of the coordinate transform.

Table 3: P-values of the Wilcoxon test comparing the model with HMC parameters tuned by maxELT & SKSD with the baseline models based on the KL-divergences of the marginals. Bold means maxELT & SKSD improves on the baseline at the given p-value.

| | Grid search | $\overline{p}_a = 0.65$ |
|---|---|---|
| $\alpha = 0$ | 0.158 | 0.627 |
| $\alpha = 1$ | **0.0102** | **0.0247** |
| ML | **0.883** | $\mathbf{2.63 \cdot 10^{-5}}$ |

and masses constant across dimension and HMC step and varying these two constants in a grid, picking the combination that gave the lowest median marginal KL-divergence to the MD training data. We also implemented another baseline, adjusting the step size constant such that the average acceptance probability was 0.65 (referred to as $\overline{p}_a = 0.65$). Further details about the implementation are given in Appendix C.1.

The marginal KL-divergences vary greatly in magnitude so we use a rank based metric to summarize the results. For each initial distribution type and marginal, we assign each HMC method a rank (1-5) according to the ordering of KL values on that marginal. We then average over the 60 marginals for each of the 15 methods, resulting in Figure 2b. For the $\alpha = 1$ trained initial distribution, maxELT & SKSD clearly results in the lowest average ranked marginal KL-divergences, meaning its samples are closest to the target. For the other two initial distributions, maxELT & SKSD is generally on-par with the best baseline. To confirm this observation, we performed a two-sided Wilcoxon test (Wilcoxon, 1945) to determine whether the differences between average rank across KL-divergences are significant or not. The resulting p-values for comparisons between maxELT & SKSD and the baselines are shown in Table 3; further details are given in Appendix C.2. The pair of models showing a significant difference, i.e., the respective p-value is smaller than 0.05, are those where our method outperforms the baseline. For the other cases, it is on par with them.

## 6 DISCUSSION AND RELATED WORK

Our experiments show a general trend that, when we solely optimize (1), the value of $\alpha$ used significantly affects performance. However, when applying the SKSD scaling, this difference becomes smaller, showing that this technique is useful for automatically finding a suitable initial distribution. In practice, it may be simple and effective to just use the standard ELBO method (i.e. $\alpha = 0$) to find an initial distribution for HMC with the SKSD scaling to ensure that it is wide enough.

Comparing to other methods for tuning the HMC hyperparameters, there are some methods also inspired by variational inference. Salimans et al. (2015); Wolf et al. (2016) use an approximation to the reverse dynamics of the chain to put a lower bound on the standard ELBO and make the optimization tractable. The method is then dependent on the accuracy of this reverse approximation with the concern that this new lower bound gets looser and looser as the the HMC chain length gets larger and larger. As our experiments in section 5.2 show, we can do just as well as these methods by directly optimizing a measure of convergence without the need for extra approximations. Caterini et al. (2018) also construct an alternative ELBO for HMC, however, they only sample the auxiliary momentum variables once at the start of the chain which reduces the empirical performance of HMC. In contrast to these methods, some gradient based tuning techniques do not use ideas from variational inference but instead optimize a proxy for mixing speed. Levy et al. (2018) generalize the standard leapfrog integrator used in HMC with multi-layer perceptrons which are then trained by maximising a modified version of the expected squared jumped distance. We improve upon this objective by directly optimizing convergence speed and using gradient information from the target distribution itself. It would be an interesting direction to use our objective to train this generalised leapfrog operator. Finally, there is the method of Titsias & Dellaportas (2019) who consider the gradient based tuning of the Markov transition operator, $p_\phi(x_t|x_{t-1})$, in the case of a single long MCMC chain. For the objective, they use the expected acceptance probability for the next step in the chain, regularized with the entropy of $p_\phi(x_t|x_{t-1})$. Unfortunately, as this entropy is intractable when using the leapfrog algorithm as the Markov transition operator, it cannot be applied to HMC in its current form.

There are also many non-gradient based heuristics for tuning the HMC hyperparameters. The popular No-U-Turn Sampler (Hoffman & Gelman, 2011) can adaptively set the number of leapfrog steps $L$ to avoid U-turns and find a global constant for the step sizes by adjusting the average acceptance

rate. In section 5.1, we found we can outperform NUTS even though we do not adaptively set L. With our objective, we can tune individual step sizes (and masses) for each dimension and step in the chain, allowing for a much higher degree of granular control over the algorithm. Furthermore, we do not need to rely on 'rules of thumb' such as standard acceptance rate targets used in many algorithms (Hoffman & Gelman, 2011; Hoffman, 2017) but we can automatically tune all continuous hyperparameters using information from the target distribution directly.

Our work also builds upon methods from statistical mechanics. The Boltzmann Generator (Noé et al., 2019) opened up this line of research by applying a normalizing flow to the problem of sampling molecular configurations. We found we can improve upon this by using the flow as the initial distribution for HMC, fine tuning the flow samples with our short chains. Our Alanine Dipeptide experiment comes from the recent work of Wu et al. (2020) on stochastic normalizing flows consisting of stochastic steps interspersed between deterministic steps in a normalizing flow. An interesting future extension would be to combine the approaches and tune parameters within the stochastic layers with our objective.

## 7 CONCLUSION

In this work, we presented a new objective motivated by the variational inference ELBO that can be easily used for the gradient-based optimization of HMC hyperparameters. We provided a fully automatic method for choosing an initial distribution for the HMC chain that reduces burn-in time and aids optimization. Evaluating on multiple real-world problems, we find our method is competitive with or improves upon existing tuning methods. We hope this encourages further work applying this idea to other methods that use HMC and that would benefit from increased convergence speed.

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

# A  2D DISTRIBUTIONS

## A.1  EQUATIONS FOR 2D TARGETS

Table 4: Unnormalized log densities for the 2D distributions used in the first experiment.

| Name | Unnormalized log density, $\log \pi^*(x)$ |
|---|---|
| Gaussian | $-\frac{1}{2}\left(\frac{32}{19}x_1^2 - \frac{60}{19}x_1 x_2 + \frac{40}{19}x_2^2\right)$ |
| Laplace | $-|x_1 - 5| - |x_2 - 5|$ |
| Dual Moon | $-3.125\left(\sqrt{x_1^2 + x_2^2} - 2\right)^2 + \log\left[\exp\left(-0.5\left(\frac{x_1+2}{0.6}\right)^2\right) + \exp\left(-0.5\left(\frac{x_1-2}{0.6}\right)^2\right)\right]$ |
| Mixture | $\log\left[\sum_{i=1}^7 \exp\left(-0.5\left[\left(x_1 - 5\cos\left(\frac{2i\pi}{7}\right)\right)^2 + \left(x_2 - 5\sin\left(\frac{2i\pi}{7}\right)\right)^2\right]\right)\right]$ |
| Wave 1 | $-0.5\left(\frac{x_2+\sin(0.5\pi x_1)}{0.4}\right)^2$ |
| Wave 2 | $\log\left[\ \exp\left(-0.5\left[\frac{x_2+\sin(0.5\pi x_1)}{0.35}\right]^2\right) + \right.$ $\left. \exp\left(-0.5\left[\frac{-x_2-\sin(0.5\pi x_1)+3\exp\left(-\frac{0.5}{0.36}(x_1-1)^2\right)}{0.35}\right]^2\right)\right]$ |
| Wave 3 | $\log\left[\ \exp\left(-0.5\left[\frac{x_2+\sin(0.5\pi x_1)}{0.4}\right]^2\right) + \right.$ $\left. \exp\left(-0.5\left[\frac{-x_2-\sin(0.5\pi x_1)+\frac{3}{1+\exp-\frac{x_1-1}{0.3}}}{0.35}\right]^2\right)\right]$ |

## A.2  COMPARISON BETWEEN $-\mathbb{E}_p\left[\log p^*(x)\right]$ AND $-\mathbb{E}_{q_\phi^{(T)}}\left[\log p^*(x)\right]$

Here, we compare $-\mathbb{E}_p\left[\log p^*(x)\right]$ with $-\mathbb{E}_{q_\phi^{(T)}}\left[\log p^*(x)\right]$ to quantify the mode seeking behaviour of the different methods. We display our results in Table 5. We see that the SKSD helps the method better match the ground truth and on some distributions e.g. Gaussian, helps prevent $-\mathbb{E}_{q_\phi^{(T)}}\left[\log p^*(x)\right]$ from underestimating $-\mathbb{E}_p\left[\log p^*(x)\right]$. This may occur with a narrow initial distribution ($\alpha = 0$) because the optimization will encourage the chains to remain in the region of high log target, artificially inflating the $\mathbb{E}_{q_\phi^{(T)}}\left[\log p^*(x)\right]$ value.

Table 5: $-\mathbb{E}_{q_\phi^{(T)}}\left[\log p^*(x)\right]$ values for the baselines and the 4 variations of our method on each of the synthetic test distributions. The ground truth value $-\mathbb{E}_p\left[\log p^*(x)\right]$ is found using rejection sampling.

| | Gaussian | Laplace | Dual Moon | Mixture | Wave 1 | Wave 2 | Wave 3 |
|---|---|---|---|---|---|---|---|
| Ground-truth | 2.8083 | 2.0075 | 0.8511 | 0.9282 | 0.4994 | -0.1703 | 0.1483 |
| $\alpha = 0$ | 2.4911 | 1.9937 | 1.0315 | 0.9216 | 0.4845 | 0.0463 | 0.1314 |
| $\alpha = 1$ | 2.7861 | 2.0431 | 2.9218 | 0.9197 | 0.4982 | -0.1050 | 0.6938 |
| SKSD & $\alpha = 0$ | 2.8390 | 2.0074 | 0.8439 | 0.9174 | 0.5142 | -0.1293 | 0.2500 |
| SKSD & $\alpha = 1$ | 2.8183 | 2.0281 | 0.7987 | 0.9198 | 0.4993 | -0.1814 | 1.2002 |
| $\min \bar{p} = 0.25$ | 3.0148 | 1.9140 | 4.1515 | 2.4894 | 1.4081 | 1.4220 | 1.0397 |
| NUTS | 2.7862 | 1.9955 | 0.7828 | 0.9379 | 0.4959 | -0.1979 | 0.0911 |

## A.3  TRAINING TIME

We provide the training times for the 2D experiments in Table 6. We note that the SKSD times are almost exactly double the non-SKSD times. This is because the majority of training time is taken up by sampling from the HMC chains. Each iteration, we use one batch of 100 HMC samples to estimate the expected log target objective and another separate batch of 100 is used to estimate the

SKSD resulting in the approximate doubling of total training time. In practice, this can be made more efficient by using the same batch of samples to estimate both objectives resulting in similar training times whether or not the SKSD is included.

Table 6: Training time for synthetic 2D problems on CPU (sec / 100 iters).

|  | Gaussian | Laplace | Dual Moon | Mixture | Wave 1 | Wave 2 | Wave 3 |
|---|---|---|---|---|---|---|---|
| $\alpha = 0$ | 22.62 | 25.33 | 33.44 | 61.17 | 21.75 | 65.91 | 65.08 |
| $\alpha = 1$ | 23.60 | 25.92 | 34.15 | 62.35 | 21.98 | 67.62 | 67.02 |
| SKSD & $\alpha = 0$ | 45.44 | 50.01 | 63.8 | 128.71 | 41.67 | 131.15 | 129.79 |
| SKSD & $\alpha = 1$ | 46.44 | 50.28 | 65.35 | 132.53 | 42.2 | 133.09 | 133.48 |
| $\min \bar{p} = 0.25$ | 4.96 | 4.16 | 3.66 | 16.83 | 2.8 | 6.78 | 7.14 |

# B    DEEP LATENT GAUSSIAN MODEL EXPERIMENTS

## B.1    DETAILS ABOUT THE SETUP

In these experiments, we set the dimension of the latent variable $z$ to be 32. For our HMC variational distribution, $q_\phi^{(T)}(z|x)$, we set $T = 30$ and use 5 leapfrog updates. We only consider training the step sizes here and leave all masses at 1. As there are multiple parameters being optimized jointly, we summarise the entire method in this section for clarity. The parameters being optimized are: $\theta$ - the decoder neural network parameters, $\phi$ - the HMC step sizes (a total of $30 \times 32 = 960$ scalar values), $\psi$ - the encoder ($q_\psi^{(0)}(z|x)$) neural network parameters and $s$ - the scale factor used to scale samples from $q_\psi^{(0)}(z|x)$. There are then 4 optimization objectives:

$$\mathcal{L}_1 = \mathbb{E}_{q_\phi^{(T)}(z|x)}\big[\log p_\theta(x, z)\big] \quad \text{Our tuning heuristic} \tag{3}$$

$$\mathcal{L}_2 = \text{SKSD}\big(q_\phi^{(T)}(z|x), p_\theta(z|x)\big) \tag{4}$$

$$\mathcal{L}_3 = \mathbb{E}_{q_\psi^{(0)}(z|x)}\big[\log p_\theta(x, z) - \log q_\psi^{(0)}(z|x)\big] \quad \text{Standard ELBO} \tag{5}$$

$$\mathcal{L}_4 = \mathbb{E}_{z_{1:k} \sim q_\psi^{(0)}(z|x)}\left[\sum_{i=1}^{k}\left(\frac{\omega^i}{\sum_{j=1}^{k}\omega^j}\right)^2 \log \omega^i\right], \ \omega^i = \frac{p_\theta(x, z^i)}{q_\psi^{(0)}(z^i|x)} \quad \text{DReG IWAE objective} \tag{6}$$

Using these 4 objectives, we then follow Algorithm 1 during training. We use $\eta_{t+1} \leftarrow \text{Adam}_{\eta_t}(\mathcal{L}_i)$ to denote one gradient step using the Adam optimizer (Kingma & Ba, 2014) maximising objective $\mathcal{L}_i$ with respect to parameter $\eta$. We note that we introduce $10^5$ pre-training steps before starting HMC optimization as these updates are very quick to do and make sure the HMC optimization has a reasonable starting point. We see that when $\alpha = 0$, we train $\psi$ by maximising (5) which is equivalent to minimising the $\alpha = 0$ divergence with the target. When $\alpha = 1$ we use (6) as the objective for $\psi$ which is the doubly reparameterized version of the IWAE objective (Tucker et al., 2019). This is equivalent to minimizing the $\alpha = 1$ divergence with the target, as shown by Hernández-Lobato (2016).For the case where we do not use the SKSD then we simply omit $s$ from our model and omit the updates to $s$ from Algorithm 1. For the VAE and IWAE baselines we simply

run the pre-training steps for the full $1.5 \times 10^5$ steps using the $\alpha = 0$ or $\alpha = 1$ updates respectively.

**Algorithm 1:** DLGM Training Algorithm

---

*Pre-training steps*;
**for** $t = 1, \ldots, 10^5$ **do**
    **if** $\alpha = 0$ **then**
        $\theta_{t+1} \leftarrow \text{Adam}_{\theta_t}(\mathcal{L}_3)$;
        $\psi_{t+1} \leftarrow \text{Adam}_{\psi_t}(\mathcal{L}_3)$;
    **end**
    **if** $\alpha = 1$ **then**
        $\theta_{t+1} \leftarrow \text{Adam}_{\theta_t}(\mathcal{L}_4)$;
        $\psi_{t+1} \leftarrow \text{Adam}_{\psi_t}(\mathcal{L}_4)$;
    **end**
**end**
*HMC-training steps*;
**for** $t = 10^5, \ldots, 1.5 \times 10^5$ **do**
    **if** $\alpha = 0$ **then**
        $\psi_{t+1} \leftarrow \text{Adam}_{\psi_t}(\mathcal{L}_3)$;
    **end**
    **if** $\alpha = 1$ **then**
        $\psi_{t+1} \leftarrow \text{Adam}_{\psi_t}(\mathcal{L}_4)$;
    **end**
    $\phi_{t+1} \leftarrow \text{Adam}_{\phi_t}(\mathcal{L}_1)$;
    $\theta_{t+1} \leftarrow \text{Adam}_{\theta_t}(\mathcal{L}_1)$;
    $s_{t+1} \leftarrow \text{Adam}_{s_t}(-\mathcal{L}_2)$;
**end**

---

## B.2 EFFECTIVENESS OF THE SCALING

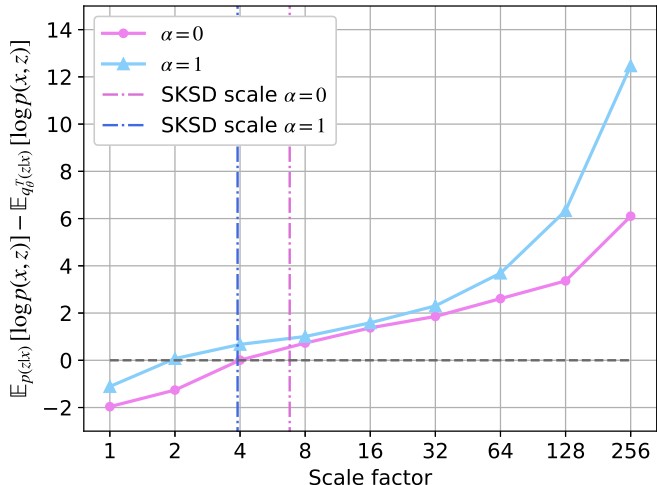

Figure 3: $\mathbb{E}_{p(z|x)}\big[\log p_\theta(x, z)\big] - \mathbb{E}_{q_\phi^{(T)}(z|x)}\big[\log p_\theta(x, z)\big]$ averaged over 200 randomly chosen MNIST test images for a range of fixed scalings used during training. The ground truth posterior $p(z|x)$ is estimated using 100 HAIS samples. The final scales found when running the training with the SKSD are also plotted.

To demonstrate the scaling's effectiveness, we run our optimization scheme on the MNIST dataset for a range of fixed scale factors and then compute $\mathbb{E}_{p(z|x)}\big[\log p_\theta(x, z)\big] - \mathbb{E}_{q_\phi^{(T)}(z|x)}\big[\log p_\theta(x, z)\big]$ with samples from $p(z|x)$ found through HAIS (Sohl-Dickstein & Culpepper, 2012). The results for

$\alpha = 0$ and $\alpha = 1$ are shown in Figure 3. For small scales, the metric is negative implying $q_\phi^{(T)}(z|x)$ is oversampling high probability regions of the target with a higher scale factor alleviating this issue. Figure 3 also shows the scale found by SKSD training when this is included in the optimization run, we see it can find an appropriate scale factor that is large enough to prevent this pathology whilst also ensuring stable performance.

## B.3 DEEP LATENT GAUSSIAN MODEL PAIRED T-TESTS

We carry out a paired t-test for each model pairing with the null hypothesis that the two population means of the log-likelihoods $\log p(x)$ are equal. Log-likelihood values are paired between models by observed data point $x$. The p-values for the tests on the MNIST dataset are given in Table 7 and the p-values for the Fashion MNIST dataset are given in Table 8. A **0** value represents that the p-value is numerically indistinguishable from 0.

Table 7: p-values for paired t-tests on test log-likelihood values on the MNIST dataset.

|  | DReG-IWAE | $\alpha = 0$ | $\alpha = 1$ | $\alpha = 0$ & SKSD | $\alpha = 1$ & SKSD | Hoffman (2017) |
|---|---|---|---|---|---|---|
| VAE | 0 | 0 | 0 | 0 | 0 | 0 |
| DReG-IWAE | - | 4.93e-18 | 0 | 0 | 0 | 0 |
| $\alpha = 0$ | - | - | 0 | 0 | 0 | 0 |
| $\alpha = 1$ | - | - | - | 1.07e-115 | 1.50e-104 | 2.04e-173 |
| $\alpha = 0$ & SKSD | - | - | - | - | 0.2405 | 1.45e-14 |
| $\alpha = 1$ & SKSD | - | - | - | - | - | 2.95e-19 |

Table 8: p-values for paired t-tests on test log-likelihood values on the Fashion MNIST dataset.

|  | DReG-IWAE | $\alpha = 0$ | $\alpha = 1$ | $\alpha = 0$ & SKSD | $\alpha = 1$ & SKSD | Hoffman (2017) |
|---|---|---|---|---|---|---|
| VAE | 0 | 0 | 0 | 0 | 0 | 0 |
| DReG-IWAE | - | 2.51e-22 | 1.25e-94 | 2.12e-205 | 0 | 4.54e-217 |
| $\alpha = 0$ | - | - | 6.46e-42 | 6.48e-124 | 0 | 5.90e-129 |
| $\alpha = 1$ | - | - | - | 8.45e-28 | 1.05e-262 | 2.04e-40 |
| $\alpha = 0$ & SKSD | - | - | - | - | 1.35e-135 | 4.81e-4 |
| $\alpha = 1$ & SKSD | - | - | - | - | - | 1.27e-79 |

## B.4 TRAINING TIME

We give the training times for the DLGM training algorithms we examined in Table 9. We first note that the VAE and IWAE baselines are much faster as they do not involve sampling from any HMC chains. We also see that the times when the SKSD loss is included are much longer than when it is omitted. This is because, in our experiments, we use a single HMC sample to estimate the expected log target objective, $\mathcal{L}_1$, at each gradient step, however, we estimate the SKSD objective, $\mathcal{L}_2$, using a further 30 samples resulting in much longer times per training step. We decided to not use the same batch of samples to estimate both $\mathcal{L}_1$ and $\mathcal{L}_2$ in our experiment in order to clearly identify the effect of the SKSD. Keeping total training iterations the same, if we had used these extra samples to estimate $\mathcal{L}_1$ then it would be unclear if any improvement was down to the SKSD or just to reduced variance in the estimate for $\mathcal{L}_1$. In practice, it would be more efficient to estimate both $\mathcal{L}_1$ and $\mathcal{L}_2$ using the same batch of HMC samples and reduce the total number of training steps as the reduced variance of $\mathcal{L}_1$ will help speed up training.

## C MOLECULAR CONFIGURATIONS SAMPLING EXPERIMENTS

### C.1 DETAILS ABOUT THE SETUP

All the RNVP models we used as initial distributions have the same architecture. They consist of five coupling blocks composed of two alternating coupling layers. The scale and shift within the

Table 9: Training Time for DLGM on GPU (sec / 1000 iters).

|  | MNIST | Fashion MNIST |
|---|---|---|
| VAE | 2.95 | 3.34 |
| DReG-IWAE | 4.34 | 4.45 |
| $\alpha = 0$ | 102.73 | 103.13 |
| $\alpha = 1$ | 103.66 | 104.02 |
| $\alpha = 0$ & SKSD | 1514.93 | 1520.27 |
| $\alpha = 1$ & SKSD | 1527.11 | 1539.52 |
| Hoffman (2017) | 126.84 | 133.13 |

coupling layers is given by fully connected networks having three layers with 128 hidden units each. Before each coupling block, we applied activation normalization (Kingma & Dhariwal, 2018).

For use in scaling the initial distribution samples, we use an empirical estimate of the normalizing flow sample mean using $10^5$ samples. To calculate the SKSD metric, to save computation, we use fixed vectors for what Gong et al. (2020) refer to as $\boldsymbol{g}_r$ instead of optimizing them. This was because we found $\boldsymbol{g}_r$ ended up very close to one-hot vectors during optimization anyway so this approximation does not make a large difference to the method.

Training the RNVP model with the $\alpha$-divergence when $\alpha = 0$ was achieved by using the reverse KL-divergence as a training objective (Papamakarios et al., 2019; Hernández-Lobato et al., 2016).

As baselines for tuning HMC parameters we used a grid search as well as training the acceptance probability. For the grid search, 25 different parameter settings were tested for each initial distribution. In each setting the step size and log mass is held constant over all HMC layers but 5 different step size constants are tested in combination with 5 different log mass constants giving 25 total combinations. The metric used to find the best combination was the median KL-divergence over all 60 marginals computed using $10^4$ samples against the $10^5$ molecular dynamics training data. When training the acceptance probability, the log mass was held at 0 for all layers and the step size was a constant for all layers with this step size constant being adjusted so that the average acceptance probability was 0.65. The constant was updated each training iteration with an update of the form $\epsilon_{t+1} = \epsilon_t - a_t(0.65 - \overline{p}_a)$, where $\overline{p}_a$ is the average acceptance probability and $a_t$ being a parameter decreasing according to the Robbins-Monro conditions (Robbins & Monro, 1951).

When testing the models, the KL-divergence was computed by first computing a kernel density estimate for each marginal using a Gaussian kernel with bandwidth chosen using Scott's rule and then finding the KL-divergence between the kernel density estimates using numerical integration.

## C.2 COORDINATE GROUPS AND MODEL COMPARISON

The coordinate transformation, introduced in Noé et al. (2019), which we used splits the feature dimensions in four different groups: 17 bond angles, 17 bond lengths, 17 dihedral angles, and 9 Cartesian coordinates, adding up to 60 dimensions in total. Due to their differing physical meaning and relevance, they follow different distributions. For each group, three sample marginals are shown in Figure 4, Figure 5, and Figure 6. We used the MacCallum lab's implementation for our coordinate transform which can be found at `https://github.com/maccallumlab/BoltzmannGenerator`.

The bond lengths and angles follow mostly unimodal, almost Gaussian, distributions which is due to the regular vibrations of the atoms within the molecule. The dihedral angles can have multiple modes while the Cartesian coordinates are quite irregular. As evaluating the KL-divergences on one-dimensional distributions is much easier than approximating it for higher dimensional distributions, we compared our models using the KL-divergences of these marginals. Figure 7 visualizes the distribution of all KL-divergences.

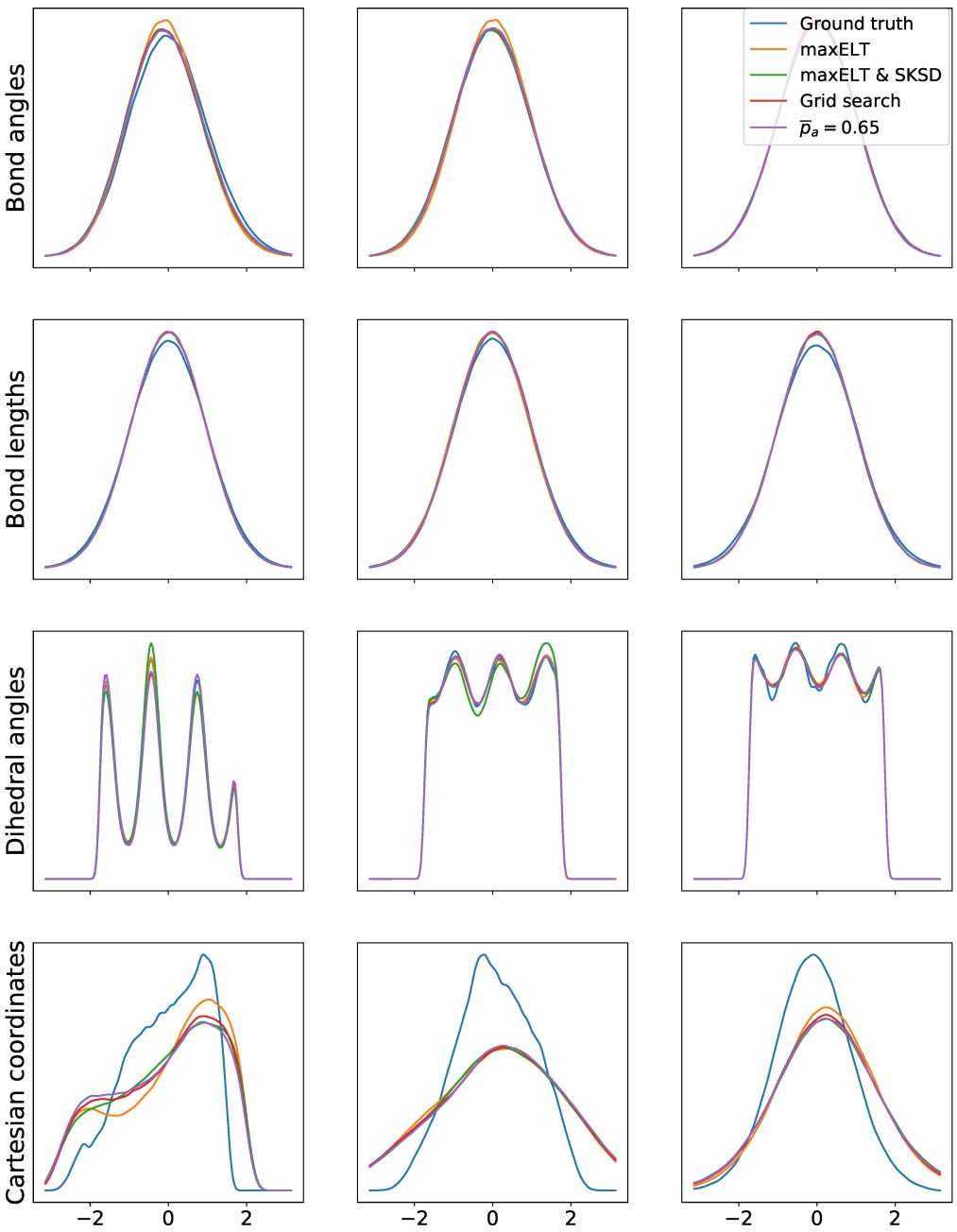

Figure 4: Sample distributions of marginals from the four coordinate groups. The graphs compare the ground truth with models having a RNVP as initial distribution followed by 50 HMC steps. The RNVP was trained with the $\alpha = 0$-divergence and the HMC parameters were tuned with the indicated methods.

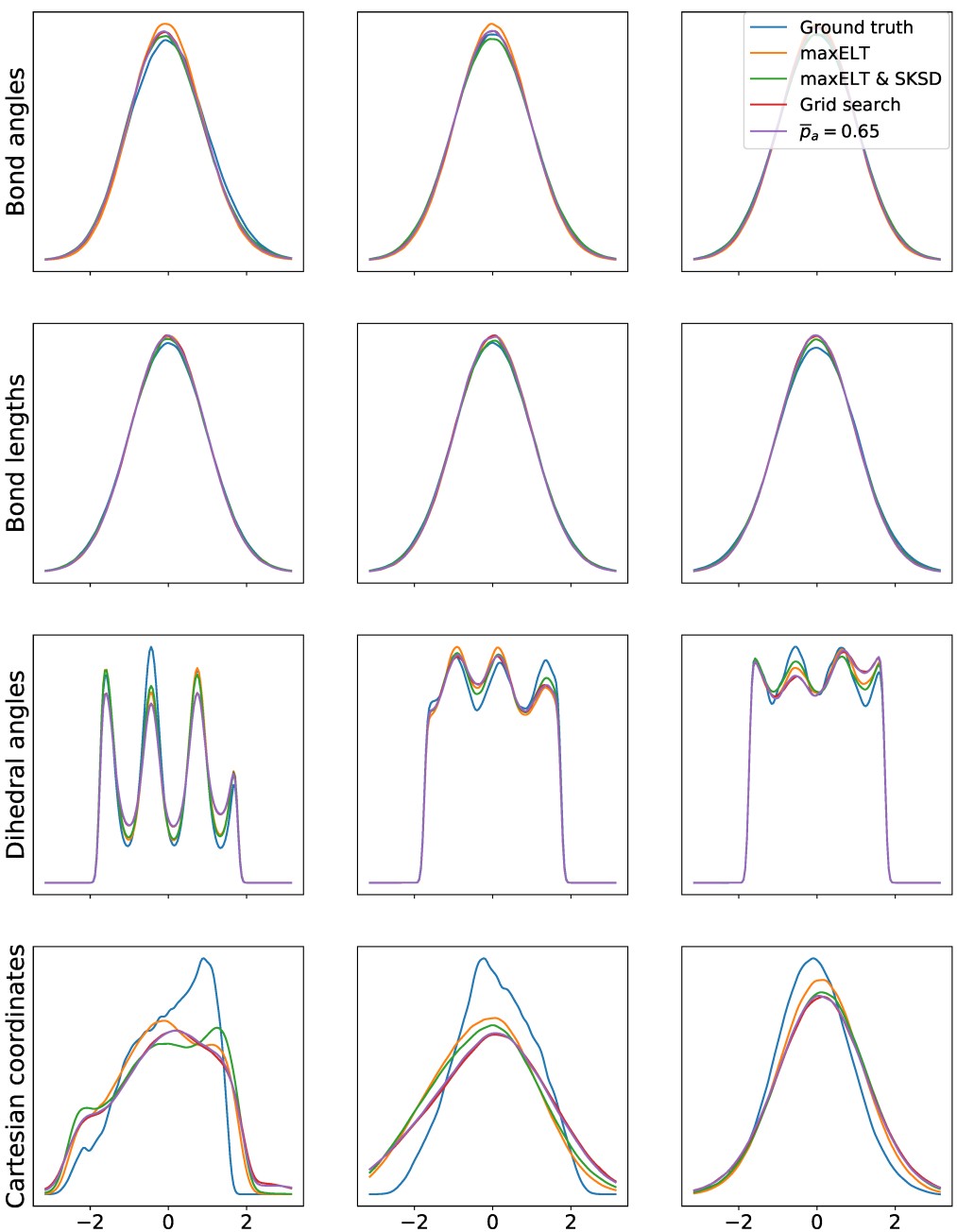

Figure 5: Sample distributions of marginals from the four coordinate groups. The graphs compare the ground truth with models having a RNVP as initial distribution followed by 50 HMC steps. The RNVP was trained with the $\alpha = 1$-divergence and the HMC parameters were tuned with the indicated methods.

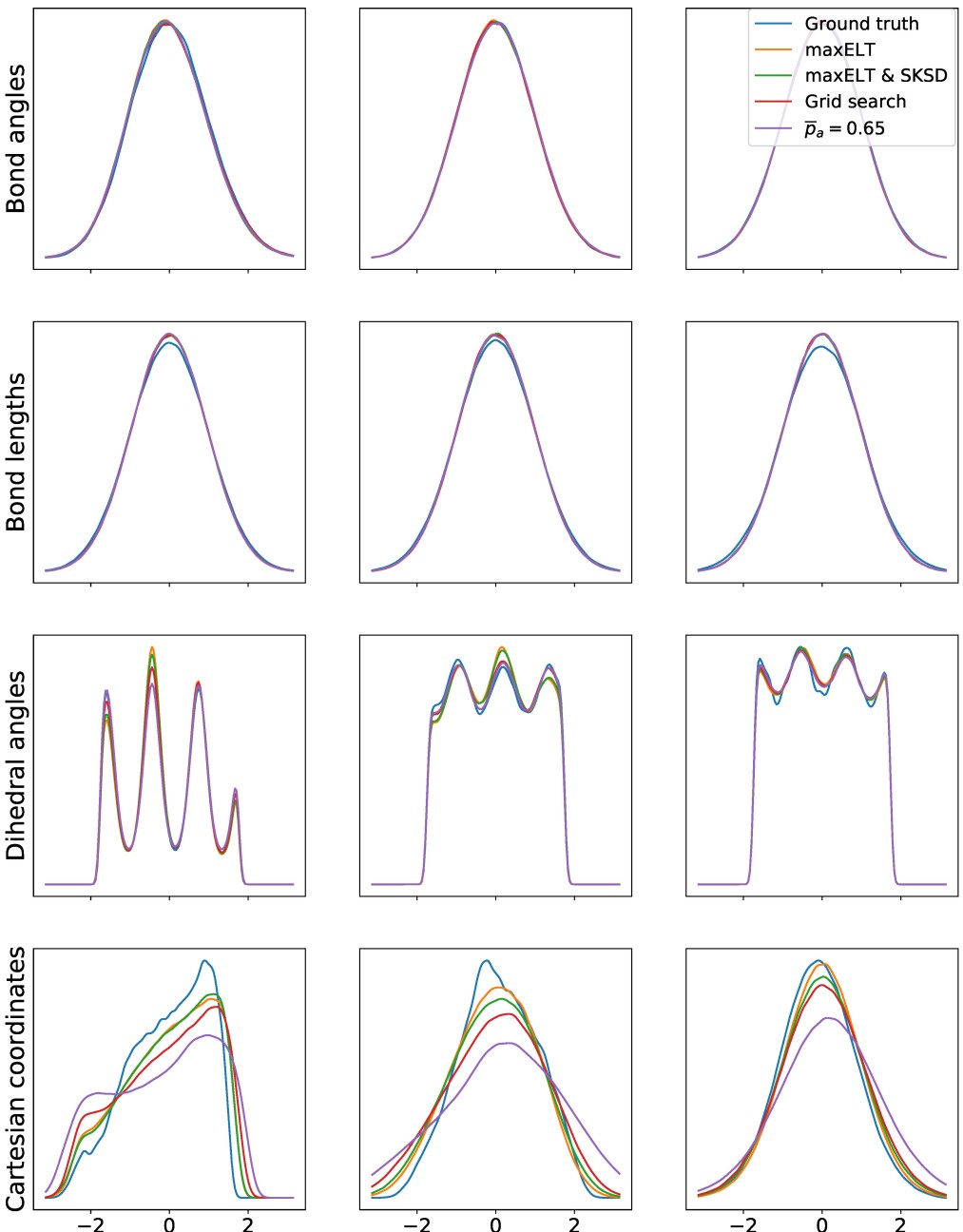

Figure 6: Sample distributions of marginals from the four coordinate groups. The graphs compare the ground truth with models having a RNVP as initial distribution followed by 50 HMC steps. The RNVP was trained via maximum likelihood and the HMC parameters were tuned with the indicated methods.

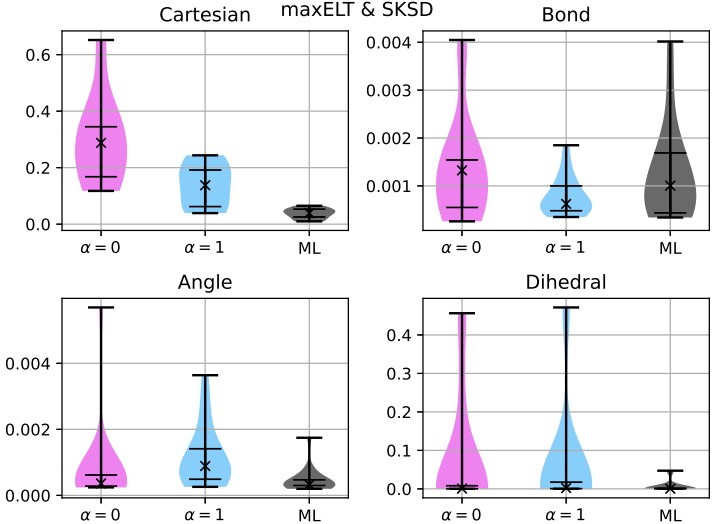

Figure 7: Violin plot of the KL-divergence of the marginals between the model tuned using maxELT & SKSD and the test set. The results are split up into the four different variable groups, i.e. Cartesian coordinates, bond lengths, bond angles, and dihedral angles. The median is marked by an x and the inner bars are the upper and lower quartiles.

Table 10: P-values of the Wilcoxon test comparing models with initial distribution trained with $\alpha = 0$-divergence. Values in **bold** mean that the test showed the top row model has lower KL-divergences than the left column model, i.e. the top row model is the better model at the significance level given by the p-value.

|  | maxELT | maxELT & SKSD | Grid search | $\overline{p}_a = 0.65$ |
|---|---|---|---|---|
| maxELT | - | **0.00056** | **0.00064** | **0.00065** |
| maxELT & SKSD | 0.00056 | - | **0.158** | **0.627** |
| Grid search | 0.00065 | 0.158 | - | 0.489 |
| $\overline{p}_a = 0.65$ | 0.00065 | 0.627 | **0.489** | - |

Clearly, there are a few outliers to both the high and the low end. To not let them distort our results, we decided to use the rank based metric to summarize our results. Furthermore, we performed a paired two-sided Wilcoxon rank test for pairs of models to assess the statistical significance of our results. The null hypothesis for this test is that the difference of the KL-divergences has a symmetric distribution around 0, i.e. there is no model with marginals consistently closer to the ground truth than the other. The resulting p-values are listed in the tables Table 10, Table 11, Table 12, and Table 13. Here, we also included an initial distribution that was trained with the $\alpha = 2$-divergence. Due to the poor performance, we did not include it in the main text. Overall, we see again that our method of tuning HMC parameters, i.e. maxELT & SKSD, is competitive or outperforms the baselines, i.e. the grid search and the training to get an average acceptance probability of 0.65.

Table 11: P-values of the Wilcoxon test comparing models with initial distribution trained with $\alpha = 1$-divergence. Values in **bold** mean that the test showed the top row model has lower KL-divergences than the left column model, i.e. the top row model is the better model at the significance level given by the p-value.

|  | maxELT | maxELT & SKSD | Grid search | $\overline{p}_a = 0.65$ |
|---|---|---|---|---|
| maxELT | - | **0.0017** | **0.024** | **0.015** |
| maxELT & SKSD | 0.0017 | - | 0.010 | 0.025 |
| Grid search | 0.024 | **0.010** | - | **0.023** |
| $\overline{p}_a = 0.65$ | 0.015 | **0.025** | 0.023 | - |

Table 12: P-values of the Wilcoxon test comparing models with initial distribution trained with $\alpha = 2$-divergence. Values in **bold** mean that the test showed the top row model has lower KL-divergences than the left column model, i.e. the top row model is the better model at the significance level given by the p-value.

|  | maxELT | maxELT & SKSD | Grid search | $\overline{p}_a = 0.65$ |
|---|---|---|---|---|
| maxELT | - | 2.4e-8 | **0.14** | **0.0066** |
| maxELT & SKSD | **2.4e-8** | - | **0.00069** | **6.6e-6** |
| Grid search | 0.14 | 0.00069 | - | **0.83** |
| $\overline{p}_a = 0.65$ | 0.0066 | 6.6e-6 | 0.83 | - |

Table 13: P-values of the Wilcoxon test comparing models with initial distribution trained with maximum likelihood. Values in **bold** mean that the test showed the top row model has lower KL-divergences than the left column model, i.e. the top row model is the better model at the significance level given by the p-value.

|  | maxELT | maxELT & SKSD | Grid search | $\overline{p}_a = 0.65$ |
|---|---|---|---|---|
| maxELT | - | **0.60** | **0.16** | 0.0094 |
| maxELT & SKSD | 0.60 | - | 0.883 | 2.6e-5 |
| Grid search | 0.16 | **0.88** | - | 6.0e-6 |
| $\overline{p}_a = 0.65$ | **0.0094** | **2.6e-5** | **6.0e-6** | - |

### C.3 SCALE PROGRESSION DURING TRAINING

To improve the overlap of the initial distribution with the target, we scale the former and learn the scaling parameter through the SKSD. Figure 8 shows the progression of the scale parameter during training for different initial distributions. The distributions trained using the $\alpha$-divergence with $\alpha = 0$ tend to be mode seeking. Hence, we expect the scale to be larger than 1 so the proposal covers the whole target distribution, and this is indeed the case. With $\alpha = 2$ the distribution tends to be mode covering, so it needs to be shrunken. The model trained with maximum likelihood already has a low KL-divergence (see e.g. Figure 2b) so there is not much modification needed, i.e. the scale is close to 1.

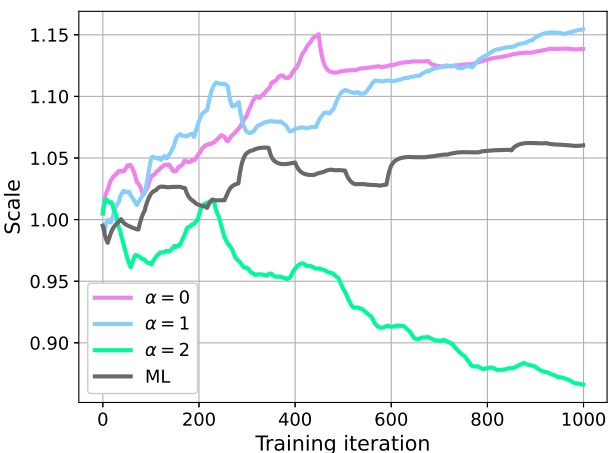

Figure 8: Progression of scale factor for maxELT & SKSD models during training. RNVP models trained by maximum likelihood and the $\alpha$-divergence with $\alpha = 0, 1, 2$ was used as initial distribution.

