# OpenReview forum: "Gradient-based tuning of Hamiltonian Monte Carlo hyperparameters"
_ICLR.cc/2021/Conference — Reject_

### Official Review · AnonReviewer2 · 2020-10-25

**Rating:** 5
**Confidence:** 4

**Review:**

Summary:
The paper introduces a gradient-based approach for tuning the step-size and the diagonal mass matrix of HMC together with the parameters of an initial distribution for the Markov chain. They suggest different objectives amenable for SGD: maximize the expected target log-density of the final state of the chain, but also an objective to ensure a somewhat ‘wide’ initial distribution. The approach is illustrated on 2-d toy models, deep latent Gaussian models on (Fashion) MNIST and molecular configurations.

Positives:
The submission suggests a practical approach for tuning HMC that remains a challenging problem. The combination of the different objectives is new as far as I am aware.  Empirical experiments are provided to justify the approach on standard benchmark problems, where it is seems to be competitive with state of the art methods, and a more extensive study on sampling molecular configurations.

Negatives:
I feel that further arguments are needed to justify why the entropy of the proposed state can be ignored when adapting the hyperparameters of the sampler. The paper argues that “Since HMC, by construction, cannot collapse to such a point mass, we argue that the entropy term can be dropped provided the initial distribution of the chain has enough coverage of the target”. I am not convinced by this: take a standard normal target, then a leapfrog-integrator with 2 steps, unit mass matrix and step size of sqrt(2) proposes deterministically from a point mass distribution and this happens everywhere on the state space. While this might be an unrealistic example, it is not clear to me how such situations can be avoided in general.
It is also not clear to me why the Sliced Kernelized Stein Discrepancy objective automatically adjusts the width of the initial distribution. In equation (4) the discrepancy is between the final state and the target and I fail to see how this relates to the width of the initial density.

Recommendations:
I vote for a weak reject at the moment. The ideas proposed in the paper are indeed interesting. However, I am not yet convinced that the objectives yield HMC kernels that explore the state space well (so the HMC proposal does not become close to deterministic/completes a U-turn so that entropy comes largely from the initial distribution which is however trained with a different objective). Also the use of the Sliced Kernelized Stein Discrepancy specifically should be better motivated. I am happy to increase my score if the authors better clarify these points.

Further comments/issues:
The authors claim in the abstract that existing approaches “optimize a tractable lower bound that is too loose to be useful in practice”. Can this be backed up more concretely? I understand that such methods (such as Thin et al., 2020) use a looser bound, but not that these types of bounds are useless in practice.
In section 3.1, how do the acceptance rates compare for the narrow vs the wide initial distribution? My intuition would be that the acceptance rates for the narrow one are smaller than for the wide one. Would it then be possible to get a better exploration even in this case by including an objective to target an acceptance rate (say increase the stepsize if the acceptance rate is above 0.65)?


Minor comments:
Is it obvious that equation (6) minimizes the 1-divergence? For k=1, is this not the standard VAE/0-divergence, while for k>1 the IWAE objective can be seen as a 0-divergence on an extended space?
What are the \gamma variables simulated from N(0,I) exactly? Are they really the momentum variables? Are the initial momentum variables not from N(0,diag(m))?
In the experiments from Section 5.1, why do you target a minimum acceptance rate of 0.25 and not an average rate of 0.65, which seems a more common choice in the adaptive MCMC literature?

---

> ### Author Response · Authors · 2020-11-14
> **Response to Reviewer 2 (part 1/2)**
>
> We thank the reviewer for their detailed feedback and constructive comments. We address the concerns below.
>
> Regarding the example given of HMC applied to a standard normal target. We would like to clarify what the reviewer means by the method proposing deterministically from a point mass target. From our calculations, we find that if the chain begins at position and momentum $x_1$ and $\nu_1$ then after one step the position will be $\sqrt{2} \nu_1$ and after a second step the position will be $-x_1$. Since both the initial momentum and position are drawn from initial distributions, they are random so across many parallel chains, we will not be drawing from a point mass distribution.
> We are unable to provide a proof that HMC can never sample from a point mass distribution but we conjecture that this is highly unlikely on practical problems.
>
> As for why the SKSD automatically adjusts the initial distribution width even though it is the discrepancy between the final state distribution and the target, we refer back to Figure 1. Consider the case where we use initial distribution 1 (pink) with a scale of 1. After training, all step sizes will be set to 0 because the optimizer has recognized that the expected log target can be maximised by staying at the initial sample points because the initial distribution is narrow and concentrated on the target mode. The final state distribution in this case will be equal to the initial distribution and we note that it is quite different to the target hence the SKSD will be high. If the scale were now to be increased to 4 then effectively we would be starting with initial distribution 2 (green). Now training will effectively optimize the step sizes resulting in a final state distribution that is very close to the target giving a very low SKSD. Therefore, the SKSD applied to the final state distribution provides a learning signal to increase the scale of the initial distribution because the SKSD can be decreased by increasing the initial scale. We hope this has alleviated the reviewer’s concerns for this section, we will add clarifying text here in an updated version of the paper.
>
> Regarding other bounds that have been used by other works such as Salimans et al., 2015 and Thin et al., 2020, what we mean by they are too loose in practice is that the bounds get looser as you add more steps in the chain prohibiting use on reasonably sized chains. They generally have a form of being equal to the standard ELBO subtract a KL term between a true and approximate distribution on variables relating to all the previous states in the chain. As the chain gets longer, the number of these variables increases making the KL term larger in general. This would then cause problems in optimization since the size of this term relative to the standard ELBO increases meaning the model just learns to fit to the approximate reverse distribution as opposed to fitting to the target as desired. Indeed Salimans et al., 2015 consider only very short chains (only 1 step for their DLGM experiment) whereas we apply our method to chains with 30 or 50 accept/reject steps. We will add text clarifying this point to the paper.
>
> When evaluating our method, we found acceptance rates to consistently stay uniformly quite high (near 1) when applied to different targets. The ability to explore comes from the granularity of control afforded to the model as the stepsizes can be tuned on a per-dimension and per-accept/reject step level as opposed to one uniform constant. Regarding an extra incentive to hit a certain target acceptance probability, in this paper we just wanted to focus on an objective inspired by variational inference without needing to use rules of thumb for targeted acceptance rates though it may be indeed an interesting further direction to look at combining our objective with a targeted acceptance rate objective.

---

> > ### Comment · AnonReviewer2 · 2020-11-22
> > **Response to authors**
> >
> > Thanks for addressing some concerns.
> >
> > What I have meant with a deterministic HMC proposal for the standard Gaussian target and step size \sqrt(2) is that for a given initial position x_1, the final state of the Markov chain after K Metropolis-Hasting steps with 2 leapfrog-steps is completely deterministic (independent of the sampled velocities).
> > For x_1 drawn from a non-deterministic initial distribution, the distribution of the position after K MH-steps will then also be non-deterministic, but will coincide with the initial distribution for even K.
> >
> > My point here was that the entropy of the HMC proposal can influence the efficiency of exploring the state space, but the entropy terms are neglected in the proposed approach for tuning HMC parameters. I agree with the authors reply to another reviewer that computing the entropy of the one step HMC-proposal is non-trivial in contrast to the more tractable MALA case as done in Titsias and Dellaportas (2019). My concern is therefore more general if the proposed objective discourages inefficiencies such as U-turns that are somehow related to the entropy – and combining it with NUTS-type strategies might be illustrating.
> >
> > The motivation for the training objective of the initial distribution via the SKDS and its automatic adjustment to the initial distribution is still not clear to me in the general case (beyond the mentioned one-dimensional Gaussian example). What happens say for a Gaussian initial distribution if the target is multimodal or has heavy tails and adapting HMC seems more challenging?

---

> > > ### Author Response · Authors · 2020-11-23
> > > **Response to Reviewer 2**
> > >
> > > Thank you for your response.
> > >
> > > We agree with the reviewer that this new objective has many features that are worth further exploration. It would be enlightening in further work to investigate the U-turn behaviour of our trained HMC chains and perhaps combine with a NUTS-type strategy. However, we feel this type of analysis may be out of the scope of our current preliminary paper, introducing this objective and getting first results on relevant problems.
> > >
> > > As for the motivation for our SKSD objective, we believe the intuition gained from the Gaussian example extends to more complex targets. In any case, if the initial distribution is too narrow and so the HMC chains cannot explore the full extent of the target, whether that be heavy tails or other modes, the SKSD will be large as it measures all discrepancy between the HMC sampling distribution and the target. Therefore, there will be a learning signal to increase the scaling in order to allow the HMC chains to fully explore the target regardless of the specific form/shape the target takes. Conversely, if the scale is too large and the HMC chains over sample from the tails, then there will also be a signal to decrease the scaling due to this discrepancy being represented in the SKSD.

---

> ### Author Response · Authors · 2020-11-14
> **Response to Reviewer 2 (part 2/2)**
>
> We address the minor comments here.
>
> It is not obvious that equation (6) minimizes the $\alpha=1$ divergence, we refer the reviewer to our reference cited in the paper for an explanation for why this is the case. The talk cited is available online at https://www.youtube.com/watch?v=Ev-6s8b3QrI please refer to 18:10 to 20:35 for the explanation. For k=1 it is indeed the standard VAE objective so k does have to be greater than 1 for the comparison with the $\alpha=1$ divergence to be valid.
>
> The $\gamma$ variables in section 3 are the sources of randomness from which the momentum variables will be derived. The function $f_\phi$ encapsulates all transformations applied to these primitive random variables including the transformation that will transform the $\gamma$ variables N(0, I) into the momentum variables N(0, diag(m)).
>
> Another reviewer has also highlighted the choice of acceptance probability so we reproduce our response here to this concern.
> The target minimum acceptance rate was chosen to be 0.25 to be in line with the original work “Learning Deep Latent Gaussian Models with Markov Chain Monte Carlo, Hoffman 2017”. In this work, HMC was used targeting the posterior in deep latent gaussian models. During training batches of training samples are taken giving a batch of posteriors to target, $\\{p(z|x_n)\\}$. The stepsize is then chosen to keep the minimum acceptance probability for any given $x_n$ to be 0.25 in order to allow the worst case chain to still mix. We agree this choice is rather strange when applied to this experiment with fixed targets, however, this was done to ensure consistency between the 2D experiments and DLGM experiments. The inclusion of the NUTS baseline ensures there is still a challenging SOTA method to compare to for this experiment.

---

### Official Review · AnonReviewer4 · 2020-10-26
**an engineering trick with limited practical significance**

**Rating:** 4
**Confidence:** 4

**Review:**

The paper proposes a method to optimize the parameters of the Hybrid Monte Carlo (HMC) algorithm (the step size and the diagonal of momentum's covariance matrix). In order to do that, the authors consider the distribution of samples q_T() obtained after T iterations of the algorithm (T accept/reject steps) starting from some distribution q_0(). Then, a reasonable objective for the optimization would be the KL-divergence between q_T() and the target density p(). However, the evaluation of the KL-divergence includes the entropy of q_T(), whose density is intractable due to numerous accept/reject steps. The proposed solution to this difficulty is to ignore the entropy term and maximize the log density of the target on samples from q_T(). To avoid the degenerate solution (due to ignorance of the entropy), the authors propose to choose q_0() carefully, e.g., to learn q_0() as a normalizing flow approximating the target p() via minimization of mass-covering alpha-divergence. The latter involves the usage of samples from the target distribution.

Major concerns:
1. The method is an engineering trick rather than a grounded approach to the optimization of sampling algorithms. Indeed, in many cases, people use MCMC methods to obtain guarantees for the sampling procedure. The proposed method removes all these guarantees by relying on the choice of the initial distribution q_0(). Moreover, the optimization of q_0() via mass-covering objectives is a notoriously hard problem since samples from the target distribution are not given in a usual setting.

2. I think the paper lacks an essential comparison with the method proposed by Titsias (Gradient-based Adaptive Markov Chain Monte Carlo, 2019). This paper proposes a more general objective for parameter optimization explicitly fostering high entropy of the proposal. Moreover, in contrast with the learning step of q_0(), it operates in an adaptive manner, not requiring any pretraining steps.

3. Given the limited theoretical novelty, I would expect the ICLR paper to demonstrate highly successful empirical results. However, it is not the case for the current submission. I'm quite confident that the results on CV tasks are out of practical interest. Also, for the molecular dynamics, the metrics' choice hinders the assessment of the practical significance.

Minor comments:
1. I don't find the comparison of marginal distributions on the 60d problem to be a convincing way to compare samplers' performance. I would suggest considering either another metric or another problem.
2. I also would suggest to include the description (at least the formula for the density) of the problem "molecular configurations." It would provide the reader with an additional intuition on its difficulty.
3. I think section 4 would benefit from the clear description of the choice of s, for instance, from the description of the variable mu, which appears there for the first time.

Additional comments:
After rereading the review, I feel that it may sound a bit harsh for the authors. Therefore, I want to say aloud that I find the paper's subject to be of great interest, consider any work in this direction valuable, and encourage the authors to continue their studies. My criticism is only an attempt to approach the review process objectively.

---

> ### Author Response · Authors · 2020-11-14
> **Response to Reviewer 4**
>
> We thank the reviewer for their detailed feedback and constructive comments. We address the concerns below.
>
> Major concerns:
>
> 1) We agree with the reviewer that our method is heuristic based, however, we are targeting the parallel MCMC use-case where many independent samples are taken from the ends of short parallel MCMC chains. Since the chains are short, very few guarantees can be given about convergence to the true target and indeed in this regime we need to use engineering methods such as well chosen initial distributions and tuned parameters to get good results. Our method is not applicable to the long chain case where such guarantees may be given. As for the reviewer’s second point regarding mass covering objectives, we agree that this is a hard problem but we must clarify that the alpha-divergence objective we use does not require any samples from the true target distribution - it only requires the target density. The wording in the paper may be confusing as we also mention training via maximum likelihood using target samples (if they are available) which is also a mass-covering objective but this is distinct to the $\alpha=1$ divergence objective which is mass-covering and doesn’t require target samples. We will clarify this in an update to the paper.
>
> 2) Regarding the paper “Gradient-based Adaptive Markov Chain Monte Carlo, Titsias 2019” we acknowledge that a reference to this work should have been made in the paper, we will add this in a new version. However, we believe it is not directly comparable to our method because the method proposed by Titsias cannot be applied to HMC in its current form as it requires the entropy of the markov one step proposal $p(x_t | x_{t-1})$ which would be intractable for HMC. Indeed, they only use their tuning method on random walk metropolis and metropolis adjusted langevin samplers.
>
> 3) Regarding the empirical results on the DLGMs, we agree that the method may not be useful for computer vision tasks in practice, however, this example was to show that our method can be competitive with state of the art methods for training DLGMs on complex data of which images is a good example rather than showing good performance on computer vision specifically. Our method could be used on other modalities in practice but many similar methods have evaluated on MNIST so we felt this was a good application to benchmark on. As for the experiment on molecular configurations, we would like to highlight our result that we can improve over grid searching for HMC parameters when using an alpha-divergence initial distribution. We believe improving over this brute force approach is a significant result regarding the usefulness of our method. We admit that the metric used is non-ideal but this is due to the high level of difficulty of this problem and this level of difficulty is precisely why we evaluate on this problem.
>
> Minor comments:
>
> 1) As we mentioned previously, the level of difficulty of this problem prevents the use of other measures of performance such as the KLD of the overall distribution due to the high dimensionality. Furthermore, the marginals of the Boltzmann distribution of proteins, especially those of the dihedral angles, are of high importance for the analysis of their properties such as how they fold, so our performance measure is of practical relevance. We will update section 5.3 in this regard. Would it be possible for the reviewer to suggest other methods that could be used to evaluate performance on this problem?
>
> 2) We will update this introduction to the molecular configurations section with additional detail in an update to the paper.
>
> 3) In section 4 we introduce the scaling factor, s, which does not need to be chosen by the user since it is automatically updated using the SKSD objective. The variable mu is the mean of the samples from the initial distribution (which is known when the initial distribution is a Gaussian but can be estimated easily if the initial distribution is a normalizing flow). We wonder if this has clarified the reviewer’s concerns for this section?
>
> We would like to thank the reviewer again for their encouragement for future work. Please let us know if there are any more concerns.

---

> > ### Comment · AnonReviewer4 · 2020-11-21
> > **response to authors**
> >
> > I have read the rebuttal and other reviews. Unfortunately, I don't think that my concerns are addressed, and, moreover, I don't see how they could possibly be addressed in a rebuttal or via minor changes in the paper.
> >
> > 1. I'm greatly confused by the authors' response since I do not understand how one could directly optimize the forward KL-divergence $D_{\text{KL}}(p||q)$ without sampling from $p$ (in the context where this KL divergence could not be evaluated analytically). At this point, I think the authors must clarify the method of optimization of the forward KL-divergence since it could greatly affect the approach.
> > 2. In practice, people don't usually apply HMC at all costs, “Gradient-based Adaptive Markov Chain Monte Carlo" proposes a very relevant approach to yours since Langevin dynamics and HMC have much in common (you can consider Langevin dynamics as a single step HMC).
> > 3. In contrast to your presentation, I consider the comparison against grid search to be weak rather than highlighting.

---

> > > ### Author Response · Authors · 2020-11-23
> > > **Response to Reviewer 4**
> > >
> > > Thank you for your response.
> > >
> > > Regarding the optimization of the initial distribution using the alpha-divergence, we provide a short summary here and refer the reviewer to https://arxiv.org/pdf/1511.03243.pdf for the full details.
> > > The alpha-divergence is a generalized divergence given by:
> > >
> > > $D_\alpha[p||q]=\frac{1}{\alpha(1-\alpha)}(1-\int p(x)^\alpha q(x)^{1-\alpha} dx)$
> > >
> > > It can be shown that in the limit as  $\alpha \rightarrow 0$, $D_\alpha[p||q] \rightarrow KL(q||p)$ and also that as $\alpha \rightarrow 1$, $D_\alpha[p||q] \rightarrow KL(p||q)$. As an outline of how this is proven: first consider the case as $\alpha \rightarrow 0$ then take the limit inside the integral and expand $p^\alpha =\text{exp}(\text{log}(p^\alpha))$ using the series expansion for $\text{exp}(x)$. Repeat this for $q(x)^{1-\alpha}$ and taking the limit as $\alpha \rightarrow 0$ gets the result. The result for $\alpha \rightarrow 1$ can be proven by reparameterizing e.g. $\gamma =1-\alpha$ and using the previous result.
> > >
> > > To optimize this divergence we write it in this form
> > >
> > > $D_\alpha[p||q]=\frac{1}{\alpha(1-\alpha)} -\frac{1}{\alpha(1-\alpha)} \int q(x) p(x)^\alpha q(x)^{-\alpha} dx=\frac{1}{\alpha(1-\alpha)} - \frac{1}{\alpha(1-\alpha)} E_{q(x)} [ (\frac{p(x)}{q(x)})^\alpha]$
> > >
> > > If we only know $p(x)$ up to a normalizing constant $p(x) = \frac{p*(x)}{Z}$ then the form becomes
> > >
> > > $D_\alpha[p||q]=\frac{1}{\alpha(1-\alpha)} - \frac{1}{\alpha(1-\alpha)} \frac{1}{Z^\alpha} E_{q(x)}[(\frac{p*(x)}{q(x)})^\alpha]$
> > >
> > > We then estimate the expectation using a Monte Carlo average with K samples from $q(x)$
> > >
> > > $D_\alpha[p||q] \approx \frac{1}{\alpha(1-\alpha)} - \frac{1}{\alpha(1-\alpha)} \frac{1}{Z^\alpha} \frac{1}{K} \sum_{k=1}^K (\frac{p*(x_k)}{q(x_k)})^\alpha$
> > >
> > > To minimize this with respect to $q$ we see that we only need to maximise the second term
> > >
> > > $\text{max} \frac{1}{\alpha(1-\alpha)} \frac{1}{Z^\alpha} \frac{1}{K} \sum_{k=1}^K (\frac{p*(x_k)}{q(x_k)})^\alpha$
> > >
> > > Equivalently we can maximise the log of this
> > >
> > > $\text{max}  -\text{log}(\alpha(1-\alpha)) - \text{log}(Z^\alpha)+\text{log}(\frac{1}{K} \sum_{k=1}^K (\frac{p*(x_k)}{q(x_k)})^\alpha)$
> > >
> > > We consider the limit as $\alpha \rightarrow 1$ from below and ignore the constant during optimization. This gives the objective we use for training.
> > >
> > > $\text{max} \quad \text{log}(\frac{1}{K} \sum_{k=1}^K \frac{p*(x_k)}{q(x_k)})$

---

### Official Review · AnonReviewer1 · 2020-10-28
**The paper is well-written, and the authors do an excellent job of articulating the problem and motivate their idea well; I do have some reservations. I vote for a weak accept.**

**Rating:** 6
**Confidence:** 3

**Review:**


### Summary:

This paper proposes a variational inference based framework to tune some of the hyper-parameters of HMC algorithms automatically. The authors drop the entropy term from regular ELBO formulation, which facilitates a gradient-based approach. However, dropping this term requires extra care for which authors offer an automated-method. Finally, the authors demonstrate empirical validation on several problems.

### Strength:

The authors do an excellent job of articulating their intuition behind the idea both (see section 3.) While dropping the entropy term from ELBO decomposition is heuristic-based, the explanations are well-formulated, and Figure 1 does an excellent job of getting the point across.

More so, since dropping the entropy term can cause pathological behaviors, the authors propose a method to ensure wider initial distributions. I commend the authors for the non-trivial engineering that was required to make their ideas work. I also, commend the author's effort of conducting statistical tests and extensive empirical evaluations.

### Concerns:

My main concern with the work is that it is often on-par with the competing methods--I understand that a new method doesn't need to be SOTA on every benchmark--and the SKSD enabled variants that achieve this performance are prohibitively slow (see Tables 6 and 9.) I could not help but feel concerned when no discussion was offered for an almost tenfold increase in the computational time for training DLGMs. To convince me, I will suggest offering an honest discussion on the run-times of the approaches.

I find the discussion in section B.1important, and believe it should be more formal.  Specifically, I will suggest algorithimizing what objective is used at which stage. Alternatively, authors can choose to restructure this some other way; however, it is too important to be left in its current form.

### Updates after the rebuttal

I like the paper and found the revised version more transparent. I support the engineering approach of the paper; however, as we all know, these papers often require authors to go to greater lengths to convince. After reading the other discussion and reviews, I think the authors can consider a few additional experiments. I would suggest investing in a more involved toy-experiment to better motivate the engineering solutions. If possible, authors can also consider a more careful ablation study to establish the relevance of each component on this toy-model. Further, the authors offered explanations for the training time aberrations; if possible, authors can consider including the equally-fast-variants in the revision to be more convincing.

---

> ### Author Response · Authors · 2020-11-14
> **Response to Reviewer 1**
>
> We would like to thank the reviewer for their comments and analysis of our paper. We address the concerns below.
>
> We acknowledge that there should have been discussion of the training time results given in the paper, we will add this text in an update to the paper. We would also like to take the chance now to explain why we ended up with these training time results. In the DLGM experiment, we were aiming at showing clearly the benefit that SKSD training can bring in terms of log-likelihood and we were not optimizing for efficiency of training. Specifically, for the case where we do not use the SKSD, for each optimization step we took only one sample from the HMC chain with which to estimate the expected log target (equation 3). For the case where we do use the SKSD, we take one HMC sample for the expected log target but then also take 30 samples to estimate the SKSD with. This will obviously take a lot longer and is an inefficient use of resources. It would be optimal to take one batch of samples e.g. 30 and then estimate both the SKSD and the expected log target with that same batch. However, if we had done that in our experiments then the variance in the expected log target estimator would have been a lot less and thus it would have been unclear if our improved results were due to the SKSD training or just due to the reduced variance in the expected log target. So in summary, our experiment using the SKSD was deliberately wasteful to make sure the benefit of the SKSD is clear.
> Indeed, in our molecular configurations experiment, we do use the same batch of samples to estimate the expected log target and the SKSD and we found that the training times are practically equivalent whether or not the SKSD term is included. We also find in the molecular configurations experiment that including the SKSD term improves performance showing that the model can be improved by adding the SKSD without increasing training time as long as the same batch of samples is used.
>
> Regarding section B.1, we will update the paper, explaining more clearly when we use each objective.

---

### Official Review · AnonReviewer3 · 2020-10-31
**interesting line of work:numerical  studies are not entirely convincing**

**Rating:** 5
**Confidence:** 5

**Review:**

Summary:
========

the article proposes to tune an HMC sampler by maximising E_\param[\log target(X_T)] over the parameters of the HMC sampler. Furthermore, the article studies the influence of the initial distribution. While the approach is certainly interesting, I have not found the empirical studies satisfying enough.

Comments:
=========
1. The article considers a vector \epsilon as well as a mass matrix. Usually, the parameter epsilon is chosen as a scalar number: choosing epsilon as a vector can indeed also be seen as a particular type of preconditioning (or choice of mass matrix). I have found this part of the paper not extremely well explained.

2. It is indeed also difficult to choose L, and that is mainly what the no-U-turn method tries to automate. In practice, dynamically adapting L can make a lot of difference in high-dimensional settings and/or different parts of the state space exhibit different scales. It would have been very interesting to investigate how the proposed method can be used **in conjunction with** no-U-turn type strategies. Furthermore, it was not entirely clear to me how the \epsilon was tuned when the no-U-turn was used.

3. In the 2D example, since the authors have used rejection sampling to produce the plots, it is also easy to accurately estimate the mean/covariance of the target distribution. It would have been interesting to use these statistics [although, it is not possible to do so in more complex scenarios] and see if this leads to improved performances.

4. in the "\min \bar{p}" method, why choose a target acceptance rate of 0.25? My experience says that the number is usually chosen much higher.

5. While reporting the KSD, I think it would have been very interesting to report the ESS [or variations of it], since it is the standard measure of efficiency in the MCMC literature.

6. Finally, while the 2D examples are certainly very interesting, I am not convinced that directly going from 2D to super-difficult-target is the right approach to understand the properties of the proposed methods. There are many settings that are more difficult than these 2D distributions, but much more tractable than the DLG/molecular targets.

In summary, I think that the authors are proposing an interesting line of research, but more careful numerical investigations are necessary to really understand the worth of the methodology.

---

> ### Author Response · Authors · 2020-11-14
> **Response to Reviewer 3**
>
> We thank the reviewer for their detailed feedback and constructive comments. We address the concerns and answer questions below.
>
> 1) We agree with the reviewer that the choice of a vector epsilon is non-standard and thus this choice should have been better explained during the introductory section of the paper. We do indeed make this choice as to allow the model more flexibility in adapting the leapfrog scheme to each specific target. We will add text clarifying this to the paper.
>
> 2) Combining our method with NUTS would be an interesting piece of future work, however, this may be out of the scope of our current paper where we attempt to provide the groundwork and initial evaluations of this new objective. With regards to our NUTS implementation, we used Algorithm 6 from “The No-U-Turn Sampler: Adaptively Setting Path Lengths in Hamiltonian Monte Carlo, Hoffman & Gelman 2011” which uses dual averaging to tune the stepsize. We will clarify that this is the case within the paper.
>
> 3) Regarding comparing the mean and covariance for the 2D toy experiments, we agree that these can also be useful for assessing varying levels of convergence to the target. However, in the main text of the paper we decided to use the kernelized stein discrepancy as this should encapsulate all discrepancy between the true distribution and sampling distribution in one number. The variance information may indeed give insightful information about any discrepancy in spread between these distributions, we can include an extra table comparing using this metric in the appendix.
>
> 4) The target minimum acceptance rate was chosen to be 0.25 to be in line with the original work “Learning Deep Latent Gaussian Models with Markov Chain Monte Carlo, Hoffman 2017”. In this work, HMC was used targeting the posterior in deep latent Gaussian models. During training batches of training samples are taken giving a batch of posteriors to target, $\\{p(z|x_n)\\}$. The stepsize is then chosen to keep the minimum acceptance probability for any given $x_n$ to be 0.25 in order to allow the worst case chain to still mix. We agree this choice is rather strange when applied to this experiment with fixed targets, however, this was done to ensure consistency between the 2D experiments and DLGM experiments. The inclusion of the NUTS baseline ensures there is still a challenging SOTA method to compare to for this experiment.
>
> 5) We would like to ask the reviewer for clarification on this point as we believe the usual ESS metric for MCMC does not apply to our method as we take samples from the ends of many short parallel chains as opposed to samples from within one long chain. Therefore, each of the samples from our method is independent and so the ESS metric would not make sense in this case.
>
> 6) In our paper, we chose challenging problems in order to stress-test the method on real-world problems from ML. We acknowledge that this may be less illuminating as to the properties of our proposed method but this does provide assurances that our method is not limited to only toy/synthetic targets. The reviewer does mention some problems that may be in between these extremes, would it be possible for the reviewer to provide some examples of problems of this type that they have in mind?

---

> > ### Author Response · Authors · 2020-11-21
> > **Follow-up regarding 2D experiments**
> >
> > We have looked at the comparison of methods using the mean and covariance statistics within the 2D experiments, however, we believe that these metrics may not be very useful in this case. Many of the distributions we look at are highly non-Gaussian so we think that the use of mean and covariance data to compare them does not make that much sense. For the two targets that are close to Gaussian (the Gaussian and Laplace targets), we compared the mean vectors and covariance matrices between the samples and the ground truth from rejection sampling using the mean squared error (just using the upper triangular matrix for the covariance matrices):
> >
> > Mean MSE:
> >
> > |       | Gaussian   | Laplace     |
> > | :------------- | :----------: | -----------: |
> > |  $\alpha=0$ | 1.18e-3   | 4.04e-5    |
> > | $\alpha=1$   | 2.66e-3 | 2.08e-4 |
> > | SKSD & $\alpha=0$ | 8.29e-6 | 5.64e-5 |
> > | SKSD & $\alpha=1$ | 2.25e-5 | 5.00e-5 |
> > | Min $p=0.25$ | 6.85e-7 | 2.00e-6 |
> > | NUTS | 4.38e-4 | 1.23e-4|
> >
> > Covariance MSE
> >
> > |       | Gaussian   | Laplace     |
> > | :------------- | :----------: | -----------: |
> > |  $\alpha=0$ | 1.17   | 9.33e-4    |
> > | $\alpha=1$   | 8.29e-3 | 1.24e-2 |
> > | SKSD & $\alpha=0$ | 1.05e-2 | 4.99e-4 |
> > | SKSD & $\alpha=1$ | 2.12e-4 | 3.29e-3 |
> > | Min $p=0.25$ | 2.11e-2 | 5.36e-2 |
> > | NUTS | 2.84e-3 | 1.90e-3|
> >
> > We found that our method does well at estimating the covariance information. For the mean estimation the baseline "Min $p=0.25$" does seem to do better, although in this case, both the baseline and our method have estimation errors that are very low, so we believe that the differences are not very meaningful here.
> >
> > Overall, we think that comparing methods using mean and variance metrics may not provide much useful information in regards to which methods are best on these 2D targets, especially when some of them are highly non-Gaussian.

---

### Decision · Program_Chairs · 2021-01-07
**Final Decision**

**Decision:**

Reject

**Comment:**

This paper proposes a tuning strategy for Hamiltonian Monte Carlo (HMC). The proposed algorithm optimizes a modified variational objective over the T step distribution of an HMC chain. The proposed scheme is evaluated experimentally.

All of the reviewers agreed that this is an important problem and that the proposed methods is promising. Unfortunately, reviewers had reservations about the empirical evaluation and the theoretical properties of the scheme. Because the evaluation of the scheme is primarily empirical, I cannot recommend acceptance of the paper in its current form.

I agree with the following specific reviewer concerns. The proposed method does not come with any particular guarantees, and particularly no guarantees regarding the effect of dropping the entropy term and using an SKSD training scheme to compensate. While guarantees are not necessary for publication, the paper should make up for this with comprehensive and convincing experiments. I agree with R1 that more careful ablation studies on toy models are needed, if nothing else to reveal the strengths and weaknesses of the proposed approach. I would also recommend a more careful discussion about the computational cost of this method and how it can be fairly compared to baselines. I don't agree that "deliberately wasteful" experiments reveal much, especially if running more realistic experiments reduces the relative impact of the proposed method.